# Revisiting Scalarization in Multi-Task Learning: A Theoretical Perspective

**Yuzheng Hu**[1†]  **Ruicheng Xian**[1⋆]  **Qilong Wu**[1⋆]  **Qiuling Fan**[2]  **Lang Yin**[1]  **Han Zhao**[1†]

[1]Department of Computer Science  [2]Department of Mathematics
University of Illinois Urbana-Champaign
{yh46,rxian2,qilong3,qiuling2,langyin2,hanzhao}@illinois.edu

## Abstract

Linear scalarization, i.e., combining all loss functions by a weighted sum, has been the default choice in the literature of multi-task learning (MTL) since its inception. In recent years, there is a surge of interest in developing Specialized Multi-Task Optimizers (SMTOs) that treat MTL as a multi-objective optimization problem. However, it remains open whether there is a fundamental advantage of SMTOs over scalarization. In fact, heated debates exist in the community comparing these two types of algorithms, mostly from an empirical perspective. To approach the above question, in this paper, we revisit scalarization from a theoretical perspective. We focus on linear MTL models and study whether scalarization is capable of *fully exploring* the Pareto front. Our findings reveal that, in contrast to recent works that claimed empirical advantages of scalarization, scalarization is inherently incapable of full exploration, especially for those Pareto optimal solutions that strike the balanced trade-offs between multiple tasks. More concretely, when the model is under-parametrized, we reveal a *multi-surface* structure of the feasible region and identify necessary and sufficient conditions for full exploration. This leads to the conclusion that scalarization is in general incapable of tracing out the Pareto front. Our theoretical results partially answer the open questions in Xin et al. (2021), and provide a more intuitive explanation on why scalarization fails beyond non-convexity. We additionally perform experiments on a real-world dataset using both scalarization and state-of-the-art SMTOs. The experimental results not only corroborate our theoretical findings, but also unveil the potential of SMTOs in finding balanced solutions, which cannot be achieved by scalarization.

## 1 Introduction

When labeled data from multiple tasks are available, multi-task learning (MTL) aims to learn a joint model for these tasks simultaneously in order to improve generalization (Caruana, 1997; Zhang and Yang, 2021; Ruder, 2017). One common approach towards this goal is learning a joint feature representation that is shared among multiple tasks, where each task then uses a separate task-specific head for its own prediction. The hope is that by sharing a joint feature representation, the effective sample size for feature learning will be larger than that for single-task learning, hence leading to better generalization across related tasks. This simple and natural idea has proven to be successful in many applications, including object segmentation (Misra et al., 2016), natural language processing (Collobert and Weston, 2008), autonomous driving (Yu et al., 2020a), to name a few.

One typical approach to learning the shared feature model from multiple tasks is called *linear scalarization*, where the learner provides a set of convex coefficients (in the form of a probability vector) to

---

[†]Corresponding author.
[⋆]Equal contribution.

37th Conference on Neural Information Processing Systems (NeurIPS 2023).

linearly combine the loss from each task and reduces the learning problem into a scalar optimization problem. Since its inception, scalarization has been the default approach for MTL (Caruana, 1997; Ruder, 2017; Zhang and Yang, 2018; Crawshaw, 2020; Zhang and Yang, 2021), mainly due to its simplicity, scalability, as well as empirical success. However, in recent years, variants of the multiple gradient descent algorithm (MGDA) (Fliege and Svaiter, 2000) have been proposed and used in MTL (Sener and Koltun, 2018; Yu et al., 2020b; Chen et al., 2020; Liu et al., 2021a,b; Zhou et al., 2022). Referred to as specialized multi-task optimizers (SMTOs), the main idea behind them is to cast the learning problem explicitly as a multi-objective (multi-criteria) optimization problem, wherein the goal is to enforce convergence to a Pareto optimal solution across tasks. In the literature, a main driving force behind these studies has been the belief that they are better than linear scalarization when *conflicting gradients* among tasks are present (Yu et al., 2020b; Liu et al., 2021a). Nevertheless, recent empirical results tell an opposite story (Kurin et al., 2022; Xin et al., 2022; Lin et al., 2022)— with proper choices of hyperparameters and regularization techniques, scalarization matches or even surpasses SMTOs. Hence, it largely remains an open problem regarding whether there is an inherent advantage of using SMTOs for MTL over scalarization, at least from a theoretical perspective.

In this paper, we revisit scalarization from a theoretical perspective. We study whether scalarization is capable of fully exploring the Pareto front in linear MTL. Technically, we aim to understand the following question on *full exploration*:

> *For every Pareto optimum $v$ on the Pareto front induced by multiple tasks, does there exist a weighted combination of task losses such that the optimal solution of the linearly scalarized objective corresponds to $v$?*

Our fundamental thesis is that, if the answer to the above question is affirmative, then there is no inherent advantage of SMTOs over scalarization on the representation level; otherwise, there might be cases where SMTOs can converge to Pareto optimal solutions that cannot be obtained via any scalarization objective. Our findings suggest that, in contrast to recent works that claimed empirical advantages of scalarization, it is inherently incapable of full exploration. Specifically, our contributions are as follows:

1. We reveal a *multi-surface* structure of the feasible region, and identify a common failure mode of scalarization, which we refer to as "gradient disagreement" (Section 3.1). This serves as an important first step towards understanding when and why scalarization fails to achieve a specific Pareto optimum.
2. We provide necessary and sufficient conditions for scalarization to fully explore the Pareto front, which lead to the conclusion that scalarization is incapable of tracing out the Pareto front in general (Section 3.2). To the best of our knowledge, we are the first to establish both necessary and sufficient conditions for full exploration in the non-convex setting.
3. We proceed to discuss implications of our theoretical results (Section 4.1), demonstrating how they answer the open questions in Xin et al. (2022), and provide a remedy solution via randomization to counteract the fundamental limitation of scalarization (Section 4.2).
4. We additionally perform experiments on a real-world dataset using both scalarization and state-of-the-art SMTOs (Section 5). The experimental results not only corroborate our theoretical findings, but also demonstrate the potential of SMTOs in finding balanced solutions.

## 2 Preliminaries

We introduce the setup of multi-task learning (MTL) and relevant notations, along with essential background on the optimal value of the scalarization objective and the concept of Pareto optimality.

**Problem setup.** Following prior works, we consider multi-task learning with $k$ regression tasks (Wu et al., 2020; Du et al., 2021). The training examples $\{(x_j, y_1^j, \cdots, y_k^j)\}_{j=1}^n$ are drawn i.i.d. from some $\mu \in \mathcal{P}(\mathcal{X} \times \mathcal{Y}_1 \times \cdots \times \mathcal{Y}_k)$, where $\mathcal{X} \subseteq \mathbb{R}^p$ is the *shared* input space and $\mathcal{Y}_i \subseteq \mathbb{R}$ is the output space for task $i$. We denote the label of task $i$ as $y_i = (y_i^1, \cdots, y_i^n)^\top \in \mathbb{R}^n$, and concatenate the inputs and labels in matrix forms: $X = [x_1, \cdots, x_n]^\top \in \mathbb{R}^{n \times p}$ and $Y = [y_1, \cdots, y_k] \in \mathbb{R}^{n \times k}$.

We focus on linear MTL, in which a linear neural network serves as the backbone for prediction. Despite its simplicity, this setting has attracted extensive research interest in the literature (Maurer, 2006; Maurer et al., 2016; Wu et al., 2020; Du et al., 2021; Tripuraneni et al., 2021). Without loss of

generality, the model that will be used for our study is a two-layer linear network[3], which consists of a shared layer $W \in \mathbb{R}^{p \times q}$ and a task-specific head $a_i \in \mathbb{R}^q$ for each task. The prediction for task $i$ on input $x$ is given by $f_i(x, W, a_i) := x^\top W a_i$. We adopt Mean Squared Error (MSE) as the loss function, and the training loss for task $i$ is given by $L_i(W, a_i) := \|XW a_i - y_i\|_2^2$.

**Notation.** We use $[k]$ to denote the index set $\{1, \cdots, k\}$. For two square matrices $M$ and $N$ with the same dimension, we write $M \succeq N$ if $M - N$ is positive semi-definite, and $M \succ N$ if $M - N$ is positive definite. We denote $\mathrm{diag}(D)$ as the diagonal matrix whose diagonal entries are given by $D = \{d_i\}_{i \in [k]}$, a set of real numbers; and $\mathrm{span}(V)$ as the linear space spanned by $V = \{v_i\}_{i \in [k]}$, a set of vectors. For two vectors with the same size, we use $\odot$ and $\oslash$ to denote entry-wise multiplication and division. When there is no ambiguity, operations on vectors (e.g., absolute value, square, square root) are performed entry-wise. For any matrix $M$, $M^\dagger$ stands for its Moore-Penrose inverse. We use $\mathrm{range}(M)$ and $\mathrm{rank}(M)$ to denote its column space and rank, respectively. The expression $\| \cdot \|$ denotes the $\ell_2$ norm of a vector, $\langle \cdot, \cdot \rangle$ the inner product of two vectors, $\| \cdot \|_F$ the Frobenius norm of a matrix, and $\rho(\cdot)$ the spectral radius of a square matrix.

**Optimal value of scalarization (restated from Wu et al. (2020)).** The objective function of scalarization is a weighted sum of losses: $\sum_{i=1}^k \lambda_i L_i(W, a_i)$. Here, the convex coefficients $\{\lambda_i\}_{i \in [k]}$ satisfy $\lambda_i \geq 0$ and $\sum_i \lambda_i = 1$, and are typically obtained via grid search (Vandenhende et al., 2021). For each task, we can decompose its loss based on the orthogonality principle (Kay, 1993)

$$L_i(W, a_i) := \|XW a_i - y_i\|_2^2 = \|XW a_i - \hat{y}_i\|^2 + \|\hat{y}_i - y_i\|^2, \tag{1}$$

where $\hat{y}_i = X(X^\top X)^\dagger X^\top y_i$ is the optimal predictor for task $i$. Denote $\Lambda = \mathrm{diag}(\{\sqrt{\lambda_i}\}_{i=1}^k)$, $A = [a_1, \cdots, a_k] \in \mathbb{R}^{q \times k}$, $\hat{Y} = [\hat{y}_1, \cdots, \hat{y}_k] \in \mathbb{R}^{n \times k}$. The scalarization objective can be compactly expressed as

$$\|(XWA - Y)\Lambda\|_F^2 = \|(XWA - \hat{Y})\Lambda\|_F^2 + \|(\hat{Y} - Y)\Lambda\|_F^2. \tag{2}$$

Throughout this paper, we assume the optimal linear predictions $\{\hat{y}_i\}_{i \in [k]}$ are linearly independent.[4] This leads to two distinct cases, depending on the width of the hidden layer $q$:

1. When $q \geq k$, the minimum value of the first term is zero, i.e., every task can simultaneously achieve its optimal loss $\|\hat{y}_i - y_i\|^2$.
2. When $q < k$, the minimum value of the first term is given by the rank-$q$ approximation. In particular, let $\hat{Y}\Lambda = \sum_{i=1}^k \sigma_i u_i v_i^\top$ be the SVD of $\hat{Y}\Lambda$. According to the Eckart–Young–Mirsky Theorem (Eckart and Young, 1936), the best rank-$q$ approximation of $\hat{Y}\Lambda$ under the Frobenius norm is given by the truncated SVD: $\hat{Y}_{q,\Lambda}\Lambda := \sum_{i=1}^q \sigma_i u_i v_i^\top$. Hence, the minimum training risk is $\|(\hat{Y}_{q,\Lambda} - \hat{Y})\Lambda\|_F^2 + \|(\hat{Y} - Y)\Lambda\|_F^2$. This regime, referred to as the *under-parameterized* regime, constitutes the main focus of our study.

**Pareto optimality.** In the context of MTL, a point $\theta^*$ (shared model parameters) is a *Pareto optimum* (PO) if there does not exist a $\theta'$ that achieves strictly better performance on at least one task and no worse performance on all other tasks compared to $\theta^*$. Formally, for a set of objectives $\{L_i\}_{i \in [k]}$ that need to be minimized, if $L_i(\theta') < L_i(\theta^*)$ for some point $\theta'$, then there must exist another task $j$ such that $L_j(\theta') > L_j(\theta^*)$. The *Pareto front* (a.k.a. Pareto frontier) is the collection of objective values $(L_1(\theta), \cdots, L_k(\theta))$ corresponding to all PO.

## 3 Linear scalarization through the lens of exploring the Pareto front

In this section, we provide a fine-grained analysis of the feasible region (formally defined in Section 3.1), revealing its multi-surface structure. We proceed to analyze whether scalarization is capable of tracing out the Pareto front, and identify necessary and sufficient conditions for full exploration.

**Reformulation.** Recall that minimizing Equation (1) is equivalent to minimizing the squared loss $\|XW a_i - \hat{y}_i\|^2$, where $\hat{y}_i = X(X^\top X)^\dagger X^\top y_i$ is the optimal linear predictor for task $i$. Additionally, the task-specific heads can be optimized independently for each task. Thus, if we let $Z = XW$ and

---

[3]Any deep linear networks for MTL can be reduced to the case of a two-layer linear network, with one layer of shared representations and one layer of task-specific heads.

[4]As long as $n \geq k$, an arbitrarily small perturbation can make $\{\hat{y}_i\}_{i \in [k]}$ linearly independent. We mainly adopt this assumption for ease of presentation.

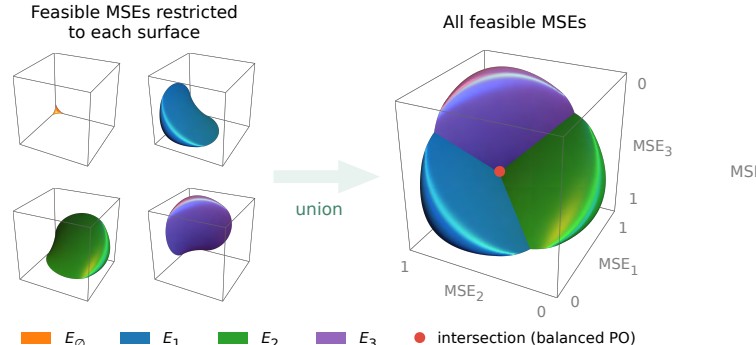

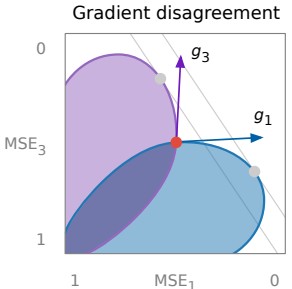

Figure 1: Feasible MSEs of a three-task linear MTL problem with $q = 1$, formed by a union of multiple feasible MSEs surfaces defined in Theorem 3.1. Note that the balanced PO located at the intersection of the surfaces is not achievable by scalarization. See Appendix A.1 for implementation details.

Figure 2: When restricted to a single surface, each contains a (different) optimum point, and the gradients (w.r.t. minimizing total MSE) on each surface disagree at their intersection.

$a_i = (Z^\top Z)^\dagger Z^\top \hat{y}_i$ (the optimal task-specific head), then minimizing Equation (1) is equivalent to

$$\min_{Z=XW} \|Z(Z^\top Z)^\dagger Z^\top \hat{y}_i - \hat{y}_i\|^2, \tag{3}$$

which can be further transformed into

$$\max_{P_Z} \hat{y}_i^\top P_Z \hat{y}_i, \quad P_Z = Z(Z^\top Z)^\dagger Z^\top.$$

Therefore, the original MTL problem can be reformulated into the following multi-objective problem

$$\max_{P_Z} \left( \hat{y}_1^\top P_Z \hat{y}_1, \cdots, \hat{y}_k^\top P_Z \hat{y}_k \right), \tag{4}$$

where $P_Z$ is a projection matrix with rank at most $q$ (i.e., hidden layer width) and maps to the column space of $X$. Each quadratic term can be further expressed as $\|P_Z \hat{y}_i\|^2$. From here it is clear (also see Section 2) that when $q \geq k$ (the over-parametrized regime), the $k$ quadratic terms can be maximized simultaneously, meaning that the network has sufficient capacity to fit the target tasks. In this case, the Pareto front reduces to a *singleton*, and can be achieved by scalarization with any choice of convex coefficients. In fact, this holds true even for general non-linear networks as long as it is wide enough, i.e., over-parametrized, and we defer more discussions to Appendix A.2.

The main focus of our study, in contrast, lies in the *under-parametrized regime* $q < k$, where various tasks compete for the network's capacity for better prediction. We comment that this regime has been the focus of previous work and demonstrated to be beneficial in MTL (Wu et al., 2020; Wang et al., 2022). We study two extreme cases within the under-parameterized regime, $q = 1$ and $q = k - 1$. By doing so, we anticipate that our results will hold for other $k$'s in between.

## 3.1 The multi-surface structure of the feasible region

The objective function of task $i$ is $\|P_Z \hat{y}_i\|^2$, i.e., the square of the projection of $\hat{y}_i$ on $P_Z$. Since we are interested in the Pareto front, we can assume w.l.o.g. that $\mathrm{rank}(P_Z) = q$ and $\mathrm{range}(P_Z) \subset \mathrm{span}(\{\hat{y}_i\}_{i \in [k]})$ (otherwise there is a loss of model capacity, and the corresponding point will no longer be a PO). Consequently, we define the feasible region as

$$\mathcal{F}_q := \left\{ \left( \hat{y}_1^\top P_Z \hat{y}_1, \cdots, \hat{y}_k^\top P_Z \hat{y}_k \right)^\top \middle| P_Z^2 = P_Z, \ \mathrm{rank}(P_Z) = q, \ \mathrm{range}(P_Z) \subset \mathrm{span}(\{\hat{y}_i\}_{i \in [k]}) \right\}. \tag{5}$$

We now provide a fine-grained characterization of $\mathcal{F}_1$ (corresponding to $q = 1$) in the following theorem, which also serves as the basis for $q = k - 1$.

**Theorem 3.1.** *Let $G = \hat{Y}^\top \hat{Y}$ and $Q = G^{-1}$. Define*

$$Q_{i_1 \cdots i_l} := D_{i_1 \cdots i_l} Q D_{i_1 \cdots i_l}, \quad 0 \leq l \leq k, \ 1 \leq i_1 < \cdots < i_l \leq k, \tag{6}$$

*where $D_{i_1 \cdots i_l}$ is a diagonal matrix with all diagonal entries equal to 1, except for the entries at positions $i_1, \cdots, i_l$ which are equal to $-1$. Consider the following surface parameterized by $v$:*

$$E_{i_1 \cdots i_l} := \left\{ v : \sqrt{v}^\top Q_{i_1 \cdots i_l} \sqrt{v} = 1 \right\}. \tag{7}$$

*We have:*

$$\mathcal{F}_1 = \bigcup_{0 \leq l \leq k} \bigcup_{1 \leq i_1 < \cdots < i_l \leq k} E_{i_1 \cdots i_l}, \tag{8}$$

*i.e., $\mathcal{F}_1$ is the union of $2^{k-1}$ surfaces in the non-negative orthant (note $E_{i_1 \cdots i_l} = E_{\overline{i_1 \cdots i_l}}$ by symmetry).*

To help the readers gain some intuitions on the *multi-surface structure* demonstrated in Theorem 3.1, we visualize the feasible region of a three-task (i.e., $k = 3$) example in Figure 1 (more details at the end of the subsection). We also provide a proof sketch below (detailed proof in Appendix A.3).

**Proof sketch of Theorem 3.1.** A rank-1 projection matrix can be expressed as $ss^\top$ with $\|s\| = 1$. The feasible region can therefore be equivalently characterized as

$$\mathcal{F}_1 = \left\{ \left( \langle \hat{y}_1, s \rangle^2, \cdots, \langle \hat{y}_k, s \rangle^2 \right)^\top \;\middle|\; \|s\| = 1, \; s \in \mathrm{span}(\{\hat{y}_i\}_{i \in [k]}) \right\}. \tag{9}$$

Define $v_i = \langle \hat{y}_i, s \rangle$ and $v = (v_1, \cdots, v_k)^\top$. We will show the set of $v$ forms the boundary of an ellipsoid determined by $Q$. As a consequence, the set of $|v|$ is the union of multiple surfaces, each corresponding to the boundary of an ellipsoid (reflection of $Q$) in the non-negative orthant. The reflection is represented by the diagonal matrix $D_{i_1 \cdots i_l}$ in the theorem statement. Finally, $\mathcal{F}_1$, which is the set of $v^2$, can be obtained by squaring the coordinates in the previous set. This finishes the proof of Theorem 3.1.

Since the orthogonal complement of a 1-dimensional subspace of $\mathrm{span}(\{\hat{y}_i\}_{i \in [k]})$ is a $(k-1)$-dimensional subspace, we immediately have the following result (proof deferred to Appendix A.4).

**Theorem 3.2.** *Let $t := (\|\hat{y}_1\|^2, \cdots, \|\hat{y}_k\|^2)^\top$. Consider the following surface parametrized by $v$:*

$$I_{i_1 \cdots i_l} = \left\{ v : \sqrt{t - v}^\top Q_{i_1 \cdots i_l} \sqrt{t - v} = 1 \right\}. \tag{10}$$

*We have:*

$$\mathcal{F}_{k-1} = \bigcup_{0 \leq l \leq k} \bigcup_{1 \leq i_1 < \cdots < i_l \leq k} I_{i_1 \cdots i_l}, \tag{11}$$

*i.e., $\mathcal{F}_{k-1}$ is the union of $2^{k-1}$ surfaces in the non-negative orthant ($I_{i_1 \cdots i_l} = I_{\overline{i_1 \cdots i_l}}$ by symmetry).*

Theorems 3.1 and 3.2 not only offer us a glimpse into the shape of the feasible region but also provide valuable insights into the fundamental limitation of linear scalarization. The key observation is that, scalarization fails to reach the *intersection* points of two (or more) surfaces when the gradients with respect to the two (or more) surfaces disagree (i.e., they lie in different directions). We refer to this phenomenon as *gradient disagreement*, and provide an illustrating example in Figure 2. Essentially, the multi-surface structure that we have revealed implies the abundance of intersection points on the feasible region $\mathcal{F}_q$. Moreover, if any of these points were to be an PO, scalarization would not be able to trace out the Pareto front (see Appendix A.5 for a detailed explanation).

We illustrate this observation using a concrete example with three tasks in Figure 1. The feasible region consists of four surfaces, among which $E_\varnothing$ is dominated by other surfaces. The intersection of $E_1, E_2, E_3$ (the red point) is a PO that achieves balanced performance across tasks, yet scalarization cannot reach this point since it lies in a valley.

### 3.2 Necessary and sufficient conditions for full exploration

In this section, we provide an in-depth analysis of the Pareto front, which is a subset of the feasible region. Concretely, we develop necessary and sufficient conditions for scalarization to fully explore the Pareto front.

We first review some important concepts that will be frequently used in this section. Denote $\mathcal{Q} := \{Q_{i_1 \cdots i_l} \mid 0 \leq l \leq k, \; 1 \leq i_1 < \cdots < i_l \leq k\}$ as the collection of matrices which define the surfaces in the previous section. Denote $\mathcal{G} := \{G_{i_1 \cdots i_l} = Q_{i_1 \cdots i_l}^{-1} \mid 0 \leq l \leq k, \; 1 \leq i_1 < \cdots < i_l \leq k\}$. Note $G_{i_1 \cdots i_l} = D_{i_1 \cdots i_l} \hat{Y}^\top \hat{Y} D_{i_1 \cdots i_l} = (\hat{Y} D_{i_1 \cdots i_l})^\top (\hat{Y} D_{i_1 \cdots i_l})$. Therefore, $\mathcal{G}$ is a collection of gram matrices which measure the *task similarity*, and each element is obtained by flipping the signs of some optimal predictors. We use $\mathcal{E}$ and $\mathcal{I}$ to denote the collection of surfaces defined in Equations (7) and (10), respectively. We also formally define a class of matrix as follows.

**Definition 3.3** (Doubly non-negative matrix). *A doubly non-negative matrix $M$ is a real positive semi-definite (PSD) matrix with non-negative entries.*

Our analysis begins with an observation that the distribution of PO across different surfaces plays a vital role in the success of scalarization. Specifically, we want to avoid the "bad" case like the one demonstrated in Figure 1, where a PO lies on the intersection of multiple surfaces. A natural solution to circumvent this issue is to force the Pareto front to lie on a *single* surface. We therefore draw our attention to this particular scenario, and provide a necessary and sufficient condition for such configuration in the following lemma.

**Lemma 3.4.** *When $q = 1$, the Pareto front of the feasible region belongs to a single surface $E^* \in \mathcal{E}$, if and only if the corresponding $G^* = (Q^*)^{-1} \in \mathcal{G}$ is doubly non-negative. We refer to the existence of such a doubly non-negative $G^*$ in $\mathcal{G}$ as `C1`.*

We outline the proof sketch below. The detailed proof can be found in Appendix B.1.

**Proof sketch of Lemma 3.4.** For the *if* part, we employ a geometric argument. Specifically, denote the acute cone formed by the optimal predictors as $\mathcal{A}$. We will show that, if $s$ (a unit vector in Equation (9)) does not belong to the dual cone $\mathcal{A}^*$, then its reflection w.r.t. the dual cone $s'$ is a better point than $s$. As a consequence, we can remove the absolute value from the proof sketch of Theorem 3.1. This further implies that the Pareto front belongs to a single surface.

For the *only if* part, we consider the points which achieve maximum value along an axis. We will show these points are PO. Finally, we will show that if C1 does not hold, then these points must belong to different surfaces in $\mathcal{E}$.

We have a similar result for $q = k − 1$. The details of the proof are deferred to Appendix B.2.

**Lemma 3.5.** *When $q = k − 1$, the Pareto front of the feasible region belongs to a single surface $I^* \in \mathcal{I}$, if and only if the corresponding $Q^* \in \mathcal{Q}$ is doubly non-negative. We refer to the existence of such a doubly non-negative $Q^*$ in $\mathcal{Q}$ as `C2`.*

Lemmas 3.4 and 3.5 provide a refined characterization of how the PO is distributed within the feasible region: unless C1 (resp. C2) holds, the Pareto front will belong to multiple surfaces in $\mathcal{E}$ (resp. $\mathcal{I}$). This easily leads to the existence of a PO on the intersection of different surfaces (see Figure 1), which will render the failure of scalarization as a consequence. It is therefore natural for us to conjecture that C1 (resp. C2) is necessary for full exploration. We make the following assumption regarding the topology of the Pareto front.

**Assumption 3.6.** *For $q = 1$ and $q = k − 1$, we assume the Pareto front is path-connected, i.e., for every pair of PO $x$ and $y$, there exists a continuous curve on the Pareto front with $x$ and $y$ being the two endpoints.*

Assumption 3.6, combined with Lemma 3.4 (resp. Lemma 3.5), will help locate a PO $P$ on the intersection of two surfaces when C1 (resp. C2) does not hold. We make an additional technical assumption to circumvent degenerated cases.

**Assumption 3.7.** *We assume $P$ (the intersection point) is a relative interior point (Rockafellar, 1997, Section 6) of the Pareto front.*

We are now ready to present the main results in this section (full proofs in Appendices B.3 and B.5).

**Theorem 3.8** (Necessity of C1 (resp. C2)). *For $q = 1$ (resp. $q = k − 1$), under Assumptions 3.6 and 3.7, C1 (resp. C2) is a necessary condition for scalarization to fully explore the Pareto front.*

**Proof sketch of Theorem 3.8.** Assumption 3.7 allows us to perform dimensionality analysis in a small neighborhood of the intersection point $P$, from which we can demonstrate the existence of a PO lying on the intersection of two surfaces with gradient disagreement. This point cannot be achieved by linear scalarization.

**Remark 3.9.** *Assumptions 3.6 and 3.7 are mainly used to simplify analysis, and they are not necessary for Theorem 3.8 to hold. As a matter of fact, we can show the necessity of C1 (resp. C2) when $k = 3$ without these assumptions. We refer the readers to Appendix B.4 for more details.*

Finally, without relying on the above assumptions, we can show that C1 and C2 are sufficient.

**Theorem 3.10** (Sufficiency of `C1` (resp. `C2`))**.** *For $q = 1$ (resp. $q = k - 1$), `C1` (resp. `C2`) is a sufficient condition for scalarization to fully explore the Pareto front.*

**Proof sketch of Theorem 3.10.** For $q = 1$, we consider the tangent plane at a PO and attempt to show it lies above the feasible region. This is transformed into demonstrating the positive semi-definiteness of a matrix, which can be conquered by applying the Perron-Frobenius theorem (Shaked-Monderer and Berman, 2021, Section 1.3) on non-negative matrices. For $q = k - 1$, we show the boundary of the feasible region is concave, and apply the supporting hyperplane theorem (Boyd et al., 2004, Section 2.5.2) to conclude the proof.

## 4 Discussion

We discuss the implications of our theoretical results, and provide a remedy solution for linear scalarization to address its inability of full exploration.

### 4.1 Implications of the theoretical results

**Fundamental limitation of linear scalarization.** From Theorem 3.8, it is clear that linear scalarization faces significant challenges in fully exploring the Pareto front in the case of $q = 1$ and $q = k - 1$, since both `C1` and `C2` require the existence of a doubly non-negative matrix, which is a very small subset of the PSD matrix. To illustrate this point, we consider a simplified probabilistic model: when the optimal predictors $\{\hat{y}_i\}_{i \in [k]}$ are drawn independently from a symmetric distribution centered at the origin, the probability that `C1` holds is $2^{-O(k^2)}$, which decays exponentially with the number of tasks. Thus, we comment that `C1` and `C2` in general does not hold in practical cases.

**Towards understanding when and why scalarization fails.** Among the vast literature that studies Pareto optimality in MTL, there is a noticeable absence of a systematic and rigorous exploration of *when and why scalarization fails to reach a Pareto optimum*. Prior work typically resorts to hypothetical examples (e.g., Figure 4.9 in (Boyd et al., 2004)) to demonstrate the failure of scalarization. Our study, which identifies the multi-surface structure of the feasible region and reveals the phenomenon of gradient disagreement in a concrete setting, serves as a pioneering step in addressing this gap in the literature.

**A closer examination of the failure mode.** For the extreme under-parametrized case ($q = 1$), on the one hand, we can prove that scalarization is able to achieve points which attain maximum value along one direction (i.e., best performance on one task). On the other hand, by examining Figure 1, we shall see that scalarization is incapable of exploring the interior points (i.e., balanced performance across all tasks). This suggests that scalarization has the tendency to *overfit* to a small fraction of tasks, and fails to strike the right balance between different tasks. We provide more empirical evidence to support this argument in Section 5.

**Criteria for practitioners.** Theorem 3.10 demonstrates that `C1` and `C2` are sufficient for scalarization to fully explore the Pareto front. As such, these conditions can serve as valuable criteria for practitioners when deciding which tasks to combine in linear MTL. Note a naive enumeration of $\mathcal{G}$ or $\mathcal{Q}$ will lead to a time complexity of $O(2^k)$. We instead provide an $O(k^2)$ algorithm checking whether `C1` holds (Algorithm 1 in Appendix C). In essence, the algorithm aims to find a set of flipping signs $\{s_i\}_{i \in [k]} \in \{+, -\}$ that ensure positive or zero correlation between all pairs in $\{s_i \hat{y}_i\}_{i \in [k]}$. It scans through the predictors in ascending order and either determines the signs $s_j$ for each optimal predictor $\hat{y}_j$ if not determined yet, or checks for conflicts between signs that are already determined. A sign can be determined when the predictors $\hat{y}_i$ and $\hat{y}_j$ are correlated. A conflict indicates the impossibility of finding such a set of flipping signs, thus falsifying `C1`. A similar algorithm can be devised for `C2` with an additional step of Cholesky decomposition beforehand, leading to an $O(k^3)$ time complexity.

**Open questions in Xin et al. (2022).** We (partially) answer the open questions in Xin et al. (2022).

`Q1`. "Is it the case that neural networks trained via a combination of scalarization and standard first-order optimization methods are not able to reach the Pareto Frontier?"

`A1`. For Linear MTL, our answer is a definitive "Yes". We demonstrate that linear scalarization with arbitrary convex coefficients cannot trace out all the Pareto optima in general. Our results are strong in the sense that they hold on a *representation* level, meaning that they are independent of the specific

optimization algorithm employed. This suggests that the weakness we have revealed in scalarization is fundamental in nature.

Q2. "Is non-convexity adding additional complexity which makes scalarization insufficient for tracing out the Pareto front?"

A2. Yes, even some mild non-convexity introduced by two-layer linear networks will render scalarization insufficient. On the other hand, we do believe the multi-surface structure that we have unveiled in Section 3.1 is both universal and fundamental, which also contributes to the observed non-convexity in practical scenarios. In essence, what we have done partly is to penetrate through the surface and gain a holistic view of the feasible region. Notably, the key obstacle in linear scalarization, as revealed by our analysis, lies in the phenomenon of *gradient disagreement*. This cannot be readily discerned by solely examining the surface.

We make two additional comments to further clarify our standing point in comparison to this work.

1. We do not aim to refute the claims in Xin et al. (2022); as a matter of fact, we consider their empirical findings highly valuable. Instead, we would like to emphasize a discrepancy between their theory and experiments. Given that we have shown the conclusion—that linear scalarization can fully explore the PF—no long holds in the non-convex setting, their theory does not account for their experiments on modern neural networks. Therefore, we urge the research community to develop a new theory elucidating the empirical efficacy of linear scalarization.

2. While the differing settings render the results from these two papers non-comparable, they collectively offer a broader perspective on the pros and cons of scalarization and SMTOs. We believe this is of great significance to both researchers and practitioners.

## 4.2 A remedy solution

Given the fundamental weakness of linear scalarization in fully exploring the Pareto front, one might wonder whether there exists any model class where linear scalarization is sufficient to fully explore the Pareto front. The answer is affirmative if we allow ourselves to use *randomized multi-task neural networks*. More specifically, given two MTL networks $f^{(0)}$ and $f^{(1)}$ over the same input and output spaces and a fixed constant $t \in [0, 1]$, we can construct a randomized MTL network as follows. Let $S \sim U(0, 1)$ be a uniform random variable over $(0, 1)$ that is independent of all the other random variables, such as the inputs. Then, consider the following randomized neural network:

$$f(x) := \begin{cases} f^{(0)}(x) & \text{if } S \leq t \\ f^{(1)}(x) & \text{otherwise} \end{cases} \tag{12}$$

Essentially, the randomized network constructed by Equation (12) will output $f^{(0)}(x)$ with probability $t$ and $f^{(1)}(x)$ with probability $1 - t$. Now if we consider the loss of $f(\cdot)$, we have

$$\begin{aligned} \mathbb{E}_\mu \mathbb{E}_S[\|f(X) - Y\|_2^2] &= \mathbb{E}_S \mathbb{E}_\mu[\|f(X) - Y\|_2^2] \\ &= t\mathbb{E}_\mu[\|f(X) - Y\|_2^2 \mid S \leq t] + (1-t)\mathbb{E}_\mu[\|f(X) - Y\|_2^2 \mid S > t] \\ &= t\mathbb{E}_\mu[\|f^{(0)}(X) - Y\|_2^2] + (1-t)\mathbb{E}_\mu[\|f^{(1)}(X) - Y\|_2^2], \end{aligned} \tag{13}$$

which means that the loss of the randomized multi-task network is a convex combination of the losses from $f^{(0)}(\cdot)$ and $f^{(1)}(\cdot)$. Geometrically, the argument above shows that randomization allows us to "convexify" the feasible region and thus the Pareto front will be convex. Now, by a standard application of the supporting hyperplane theorem (Boyd et al., 2004, Section 2.5.2) we can see that every point on the Pareto front can be realized via linear scalarization. We empirically verify the effectiveness of randomization in Appendix D.3.

We comment that randomization is a powerful tool that has wide applicability in machine learning. For instance, it has achieved great success in online learning, specifically in the Follow-the-Perturbed-Leader algorithm (Kalai and Vempala, 2005). It also appears in the literature of fairness to achieve the optimal regressor or classifier (Zhao and Gordon, 2022; Xian et al., 2023). Here we further demonstrate that it can be applied to MTL to facilitate the exploration of linear scalarization. In essence, randomization increases the representation power of the original neural network.

# 5 Experiments

In this section, we corroborate our theoretical findings with a linear MTL experiment, where we show that linear scalarization fails to fully explore the Pareto front on a three-task problem that does not satisfy C1 with $q = 1$. In comparison, SMTOs can achieve balanced Pareto optimum solutions that are not achievable by scalarization.

**Dataset and setup.** We use the SARCOS dataset for our experiment (Vijayakumar and Schaal, 2000), where the problem is to predict the torque of seven robot arms given inputs that consist of the position, velocity, and acceleration of the respective arms. For ease of visualization, we restrict the tasks to arms 3, 4, and 5; in particular, they do not satisfy C1.

Our regression model is a two-layer linear network with hidden size $q = 1$ (no bias). To explore the portion of the Pareto front achievable by linear scalarization, we fit 100,000 linear regressors with randomly sampled convex coefficients and record their performance. These are compared to the solutions found by SMTOs, for which we use MGDA (Désidéri, 2012) and MGDA-UB (Sener and Koltun, 2018). Specifically, to guarantee their Pareto optimality, we run the SMTOs with full-batch gradient descent till convergence (hence each SMTO method has one deterministic solution). Additional details, including hyperparameter settings, can be found in Appendix D.1.

**Results.** In Figure 3, we plot the MSEs achieved by scalarization under randomly sampled weights. We observe that the solutions achieved by scalarization are concentrated near the vertices and partly along the edges, indicating their tendency to overfit to 1–2 of the tasks. Scalarization also fails to explore a large portion of the Pareto front in the center—which have more balanced performance across task; the inability of full exploration corroborates the necessary condition (Theorem 3.8) discussed in Section 3. In contrast, SMTOs are capable of finding solutions not achievable by scalarization that are located in the center of the Pareto front. This showcases the relative advantage and practical utility of SMTOs in MTL over scalarization, especially in scenarios where equitable performance on all tasks are desired. To strengthen our results, we experiment with multiple random initializations and observe consistent results (see Appendix D.2).

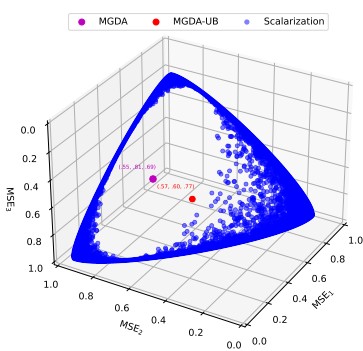

Figure 3: Linear scalarization has the tendency to overfit to a subset of tasks by finding *peripheral* blue points of the Pareto front. MGDA (magenta) and MGDA-UB (red) converge to more balanced PO in the *interior*.

**Discussion.** We would like to note that we are not endorsing any specific SMTO algorithm, such as MGDA or MGDA-UB. As a matter of fact, we hypothesize that no single algorithm is universally effective (also see (Chen et al., 2023)), notably in overcoming the limitation of scalarization in terms of full exploration. Instead, by revealing the limitation of scalarization and the potential benefits of *some* SMTO algorithms, we aim to strengthen SMTO research, challenge the notion that linear scalarization is sufficient, and advocate for a more balanced progression in the field of MTL.

# 6 Limitations and future directions

**Limitations.** There are two major limitations of our work. First, while the techniques we develop in this paper are highly non-trivial, we only cover the two extreme cases $q = 1$ and $q = k - 1$ in the under-parametrized regime, and the analysis for general $q < k$ is missing. Second, the setting of our study is relatively restricted. Specifically, we focus on linear MTL where different tasks share the same input. A more realistic setting should factor in different inputs for different tasks (Wu et al., 2020), as well as some non-linearity in the model architecture (Kurin et al., 2022).

**Future directions.** We expect the geometry of the feasible region to transition smoothly as we increase $q$, meaning that it will not abruptly collapse into a *single* surface. Providing a rigorous basis for this assertion would facilitate the analysis for general $q$'s, thereby complementing our study. Regarding the assumption of shared input, while our analysis persists under the 'proportional covariance assumption' adopted by Wu et al. (2020), we find this assumption excessively restrictive,

and hope future work could relax it further. The generalization to neural networks with ReLU activation is interesting but also challenging, and it is unclear whether the techniques and results developed in this paper can be extended to the non-linear setting. Another interesting avenue to explore is interpreting the multi-surface structure we have uncovered, which we hypothesize to have connections with game theory. Finally, our paper dedicates to demonstrating the weaknesses of linear scalarization, while we consider it an important future direction to theoretically analyze the advantage of SMTOs, as partially revealed in our experiments. In all, we hope our study can initiate a line of works on MTL, which provide theoretical justifications for the usage of specific algorithms beyond empirical evaluations across various settings.

## 7 Related work

**SMTOs.** There is a surge of interest in developing specialized multi-task optimizers (SMTOs) in the literature. One of the earliest and most notable SMTOs is the multiple-gradient descent algorithm (MGDA) (Désidéri, 2012). For each task, MGDA solves an optimization problem and performs updates using the gradient of the objective function with both shared and task-specific parameters. However, MGDA cannot scale to high dimension (e.g., neural networks), and computing the gradients for every task is prohibitively expensive. To counter these drawbacks, Sener and Koltun (2018) introduce a scalable Frank-Wolfe-based optimizer which is more efficient. Another prominent issue in MTL is that the gradients from multiple tasks could conflict with each other and the performance of some tasks could be significantly affected. To this end, Yu et al. (2020b) propose *gradient surgery* to mitigate the negative interference among tasks; Chen et al. (2020) and Liu et al. (2021a) propose to manipulate the gradients or the gradient selection procedures; Fernando et al. (2022) devise a stochastic gradient correction method and prove its convergence. More recently, Navon et al. (2022) cast the gradient update as a Nash bargaining game, yielding well-balanced solutions across the PF.

**Empirical comparison of SMTOs and scalarization.** A recent line of works empirically compare SMTOs and linear scalarization. Their central claim is that the performance of linear scalarization is comparable with the state-of-the-art SMTOs despite the added complexity of SMTOs. Xin et al. (2022) argue that multi-task optimization methods do not yield any improvement beyond what can be achieved by linear scalarization with carefully-tuned hyperparameters; Kurin et al. (2022) show that the performance of linear scalarization matches or even surpasses the more complex SMTOs when combined with standard regularization techniques. In addition to classification tasks, Vandenhende et al. (2021) conduct extensive experiments on dense prediction tasks, i.e., tasks that produce pixel-level predictions. They conclude that avoiding gradient competition among tasks can actually lead to performance degradation, and linear scalarization with fixed weights outperforms some SMTOs.

**Exploration of the Pareto front.** It is well-known that minimizing scalarization objectives with positive coefficients yields Pareto optimal solutions (Boyd et al., 2004), while using non-negative coefficients yields weak Pareto optimal ones (Zadeh, 1963). On the other hand, exploration of the *entire* Pareto front has been a long-standing and difficult open problem in MTL. For the simplest case, if all loss functions are convex or the achievable region is a convex set, then scalarization can fully explore the PF (Boyd et al., 2004; Xin et al., 2022). A slightly weaker condition for full exploration is directional convexity (Holtzman and Halkin, 1966), established by Lin (1976). Several recent works in MTL develop algorithms to generate PO subject to user preferences, in particular on trade-offs among certain tasks (Lin et al., 2019; Mahapatra and Rajan, 2020; Momma et al., 2022). There is also a line of empirical works that focuses on designing algorithms to efficiently explore the PF (Ma et al., 2020; Lin et al., 2020; Liu et al., 2021c; Navon et al., 2021; Ruchte and Grabocka, 2021; Ye and Liu, 2022). As far as we know, we are the first to establish both necessary and sufficient conditions for full exploration in the non-convex setting.

## 8 Conclusion

In this paper, we revisit linear scalarization from a theoretical perspective. In contrast to recent works that claimed consistent empirical advantages of scalarization, we show its inherent limitation in fully exploring the Pareto front. Specifically, we reveal a multi-surface structure of the feasible region, and identify necessary and sufficient conditions for full exploration in the under-parametrized regime. We also discuss several implications of our theoretical results. Experiments on a real-world dataset corroborate our theoretical findings, and suggest the benefit of SMTOs in finding balanced solutions.

## Acknowledgement

We thank the anonymous reviewers for their constructive feedback. Yuzheng Hu would like to thank Fan Wu for useful suggestions in writing. Han Zhao's work was partly supported by the Defense Advanced Research Projects Agency (DARPA) under Cooperative Agreement Number: HR00112320012, an IBM-IL Discovery Accelerator Institute research award, a Facebook Research Award, a research grant from the Amazon-Illinois Center on AI for Interactive Conversational Experiences, and Amazon AWS Cloud Credit.

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

# A    Omitted details from Section 3.1

## A.1    Implementation of Figure 1

Figure 1 is generated from a simple three-task linear MTL problem that we constructed, using Equation (4). Specifically, we set $\hat{y}_1 \approx (0.98, 0, 0.2), \hat{y}_2 \approx (-0.49, -0.85, 0.2), \hat{y}_3 \approx (-0.49, 0.85, 0.2)$ (the number of data points is $n = 3$; this is a rotated version of the equiangular tight frame), set $q = 1$ (the width of the network is one, i.e., under-parameterized), and plotted the achievable points of Equation (4) by sweeping $P_Z$ (the set of rank-1 projection matrices). The software we used is Mathematica.

## A.2    Over-parametrization reduces the Pareto front to a singleton

For general non-linear multi-task neural networks, we will show that increasing the width reduces the Pareto front to a singleton, where all tasks simultaneously achieve zero training loss. As a consequence, linear scalarization with any convex coefficients will be able to achieve this point.

To illustrate this point, we follow the same setting as in Section 2, except changing the model to be a two-layer ReLU multi-task network with bias terms. Concretely, for task $i$, the prediction on input $x$ is given by $f_i(x, W, b, a_i) = a_i^\top \max(Wx + b, 0)$. We have the following result.

**Theorem A.1.** *There exists a two-layer ReLU multi-task network with hidden width $q = nk$ and parameters $(W, b, a_1, \cdots, a_k)$, such that*

$$f_i(x_i^j, W, b, a_i) = y_i^j, \quad \forall j \in [n], \forall i \in [k]. \tag{14}$$

*This implies that the network achieves zero training loss on each task.*

*Proof.* For every $i \in [k]$, we can apply Theorem 1 in (Zhang et al., 2021) and find $(W_i, b_i, \tilde{a}_i)$ such that

$$\tilde{a}_i^\top \max(W_i x_i^j + b_i, 0) = y_i^j, \quad \forall j \in [n]. \tag{15}$$

Here, $W_i \in \mathbb{R}^{n \times p}$ and $b_i, \tilde{a}_i \in \mathbb{R}^n$. Now consider

$$W = \begin{pmatrix} W_1 \\ W_2 \\ \vdots \\ W_n \end{pmatrix} \in \mathbb{R}^{nk \times p}, \quad b = \begin{pmatrix} b_1 \\ b_2 \\ \vdots \\ b_n \end{pmatrix} \in \mathbb{R}^{nk}, \quad a_i = e_i \otimes \tilde{a}_i \in \mathbb{R}^{nk} \quad \forall i \in [n], \tag{16}$$

where $e_i$ denotes the $i$-th unit vector in $\mathbb{R}^n$ and $\otimes$ stands for the Kronecker product. It is straightforward to see that

$$a_i^\top \max(W_i x_i^j + b_i, 0) = y_i^j, \quad \forall j \in [n], \forall i \in [k]. \tag{17}$$

This finishes the proof as desired. $\square$

## A.3    Proof of Theorem 3.1

*Proof of Theorem 3.1.* Define $v_i = \langle \hat{y}_i, s \rangle$ and $v = (v_1, \cdots, v_k)^\top$. We are interested in the set

$$S := \left\{ v \mid \|s\| = 1, s \in \mathrm{span}(\{\hat{y}_i\}_{i \in [k]}) \right\}. \tag{18}$$

We will show $S$ is equivalent to the following set

$$B := \left\{ v \mid v^\top Q v = 1 \right\}, \quad Q = (\hat{Y}^\top \hat{Y})^{-1}, \tag{19}$$

which is essentially the boundary of an ellipsoid.

**Step 1:** $S \subset B$.    Note $v = \hat{Y}^\top s$, so it suffices to show

$$s^\top \hat{Y} (\hat{Y}^\top \hat{Y})^{-1} \hat{Y}^\top s = 1. \tag{20}$$

Denote $P_{\hat{Y}} = \hat{Y} (\hat{Y}^\top \hat{Y})^{-1} \hat{Y}^\top$, which is a projection matrix that maps to the column space of $\hat{Y}$. Since $s \in \mathrm{span}(\{\hat{y}_i\}_{i \in [k]})$, we have

$$s^\top P_{\hat{Y}} s = s^\top s = \|s\|^2 = 1. \tag{21}$$

**Step 2: $B \subset S$.** Since $\hat{Y}^\top$ has full row rank, for every $v \in \mathbb{R}^k$, we can always find $s \in \mathbb{R}^n$ such that $\hat{Y}^\top s = v$. We can further assume $s \in \text{span}(\{\hat{y}_i\}_{i \in [k]})$ by removing the component in the null space of $\hat{Y}^\top$. Assuming $v \in B$, we have

$$s^\top P_{\hat{Y}} s = 1, \quad s \in \text{span}(\{\hat{y}_i\}_{i \in [k]}). \tag{22}$$

Since $P_{\hat{Y}}$ is a projection matrix that maps to the column space of $\hat{Y}$, we have $s^\top s = 1$, which implies $\|s\| = 1$.

Now we have $S = B$. The set

$$S' := \left\{ |v| \mid \|s\| = 1, s \in \text{span}(\{\hat{y}_i\}_{i \in [k]}) \right\} \tag{23}$$

can be obtained by reflecting $E$ to the non-negative orthant. Formally, a reflection is determined by a collection of axes $\{i_1, \cdots, i_l\}$ ($0 \leq l \leq k$). The image of such reflection in the non-negative orthant is given by

$$B_{i_1 \cdots i_l} = \{v \mid v^\top Q_{i_1 \cdots i_l} v = 1, v \geq 0\}, \tag{24}$$

where $Q_{i_1, \cdots i_l} = D_{i_1 \cdots i_l} Q D_{i_1 \cdots i_l}$. As a consequence, we have

$$S' = \bigcup_{0 \leq l \leq k} \bigcup_{1 \leq i_1 < \cdots < i_l \leq k} B_{i_1 \cdots i_l}. \tag{25}$$

Finally, the feasible region can be equivalently characterized as

$$\mathcal{F}_1 = \left\{ v^2 \mid \|s\| = 1, s \in \text{span}(\{\hat{y}_i\}_{i \in [k]}) \right\}, \tag{26}$$

which can be obtained by squaring the coordinates of $S'$. Therefore, if we denote

$$E_{i_1 \cdots i_l} = \{v \mid \sqrt{v}^\top Q_{i_1 \cdots i_l} \sqrt{v} = 1\}, \tag{27}$$

then (note the square root naturally implies $v \geq 0$)

$$\mathcal{F}_1 = \bigcup_{0 \leq l \leq k} \bigcup_{1 \leq i_1 < \cdots < i_l \leq k} E_{i_1 \cdots i_l}. \tag{28}$$

This finishes the proof as desired. $\qquad\square$

### A.4 Proof of Theorem 3.2

*Proof of Theorem 3.2.* Note the orthogonal complement of a 1-dimensional subspace of $\text{span}(\{\hat{y}_i\}_{i \in [k]})$ is a $(k-1)$-dimensional subspace, and that the objective function for each task is the square of the projection of $\hat{y}_i$. Denote $t := (\|\hat{y}_1\|^2, \cdots, \|\hat{y}_k\|^2)^\top$. By the Pythagorean theorem (Bhatia, 2013, Section I.6), if $v \in \mathcal{F}_1$, then we must have $t - v \in \mathcal{F}_{q-1}$, and vice versa. As a consequence, denote

$$I_{i_1 \cdots i_l} = \{v \mid \sqrt{t-v}^\top Q_{i_1 \cdots i_l} \sqrt{t-v} = 1\}, \tag{29}$$

and we have

$$\mathcal{F}_{q-1} = \bigcup_{0 \leq l \leq k} \bigcup_{1 \leq i_1 < \cdots < i_l \leq k} I_{i_1 \cdots i_l} \tag{30}$$

by Theorem 3.1. $\qquad\square$

### A.5 Why scalarization fails in the presence of gradient disagreement

We explain why linear scalarization is incapable of exploring an intersection point with gradient disagreement.

We first recall the geometric interpretation (see Figure 4.9 in (Boyd et al., 2004)) of linear scalarization: a point $P$ lying at the boundary of the feasible region can be achieved by scalarization, if and only if there exists a hyperplane at $P$ and the feasible region lies above the hyperplane. The normal vector of the hyperplane is proportional to the scalarization coefficients. By definition, if a hyperplane at $P$ lies below the feasible region, its normal vector must be a subgradient of the surface. When the boundary of the feasible region is differentiable and the subdifferential set is non-empty, the normal vector must be the gradient of the surface, and the hyperplane becomes the tangent plane at $P$.

Now suppose $P$ lies at the intersection of two differentiable surfaces $S_1$ and $S_2$, and that $P$ can be achieved by scalarization. Applying the above argument to $S_1$ and $P$, we know that the scalarization coefficients are proportional to the gradient w.r.t. $S_1$. Similarly, applying the above argument to $S_2$ and $P$ yields that the scalarization coefficients are proportional to the gradient w.r.t. $S_2$. This will result in a contradiction if the two gradients w.r.t. $S_1$ and $S_2$ at $P$ lie in different directions, a phenomenon we refer to as *gradient disagreement*. In this case, scalarization cannot reach $P$.

## B  Omitted details from Section 3.2

### B.1  Proof of Lemma 3.4

We begin by reviewing some basics of convex analysis. We refer the readers to Rockafellar (1997) for more background knowledge.

**Definition B.1** (Convex hull). *Let $X = \{x_i\}_{i \in [m]}$ be a collection of points in $\mathbb{R}^n$. The **convex hull** of $X$, denoted as $\mathrm{conv}(X)$, is the set of all convex combinations of points in $X$, i.e.,*

$$\mathrm{conv}(X) := \left\{ \sum_{i \in [m]} \alpha_i x_i \mid \alpha_i \geq 0, \sum_{i \in [m]} \alpha_i = 1 \right\}. \tag{31}$$

**Definition B.2** (Cone and convex cone). *A set $K \subset \mathbb{R}^n$ is called a **cone** if $x \in K$ implies $\alpha x \in K$ for all $\alpha > 0$. A **convex cone** is a cone that is closed under addition. The convex cone generated by $X = \{x_i\}_{i \in [m]} \subset \mathbb{R}^n$, denoted as $\mathrm{cone}(X)$, is given by*

$$\mathrm{cone}(X) := \mathrm{conv}(\mathrm{ray}(X)), \tag{32}$$

*where $\mathrm{ray}(X) = \{\lambda x_i \mid \lambda \geq 0, i \in [m]\}$.*

**Definition B.3** (Dual cone). *The **dual cone** of $C \subset \mathbb{R}^n$, denoted as $C^*$, is given by*

$$C^* := \{y \mid \langle y, x \rangle \geq 0, \ \forall x \in C\} \tag{33}$$

**Definition B.4** (Projection onto a convex set). *Let $C$ be a closed convex set in $\mathbb{R}^n$. The **orthogonal projection** of $y \in \mathbb{R}^n$ onto the set $C$, denoted as $P_C y$, is the solution to the optimization problem*

$$P_C y := \inf_{x \in C} \{\|x - y\|\}. \tag{34}$$

**Theorem B.5** (Bourbaki-Cheney-Goldstein inequality (Ciarlet, 2013)). *Let $C$ be a closed convex set in $\mathbb{R}^n$. For any $x \in \mathbb{R}^n$, we have*

$$\langle x - P_C x, y - P_C x \rangle \leq 0, \quad \forall y \in C. \tag{35}$$

Theorem Theorem B.5 is also known as the variational characterization of convex projection. We will use it to prove the following lemma, which serves as a crucial ingredient in the proof of Lemma 3.4.

**Lemma B.6.** *Let $C$ be a closed convex set in $\mathbb{R}^n$. For any $z \in \mathbb{R}^n$, we have*

$$\langle z', x \rangle \geq \langle z, x \rangle, \quad \forall x \in C, \tag{36}$$

*where $z' = 2P_C z - z$ is the reflection of $z$ w.r.t. $C$.*

*Proof.* Plugging in $z' = 2P_C z - z$, it suffices to show

$$\langle P_C z - z, x \rangle \geq 0, \quad \forall x \in C. \tag{37}$$

This is true because

$$\langle P_C z - z, x - P_C z \rangle \geq 0, \quad \forall x \in C \tag{38}$$

by Theorem B.5, and that

$$\langle z - P_C z, P_C z \rangle = 0. \tag{39}$$

$\square$

We are now ready to prove Lemma 3.4.

*Proof of Lemma 3.4.* For the *if* part, we assume w.l.o.g. that $G = \hat{Y}^\top \hat{Y}$ is doubly non-negative. This implies that the angle between each pair of optimal predictors is non-obtuse. Denote $\mathcal{A} = \mathrm{cone}(\{\hat{y}_i\}_{i \in [k]})$ and $\mathcal{A}^*$ as its dual cone. We have $\mathcal{A} \subset \mathcal{A}^*$.

Our goal is to show that, for every $\|s\| = 1$, we can always find $s' \in \mathcal{A}^*$, such that 1) $\|s'\| = 1$; 2) $\langle s', \hat{y}_i \rangle \geq |\langle s, \hat{y}_i \rangle|, \forall i \in [k]$. This implies that when restricting our discussion to the Pareto front, we can safely ignore the absolute value which appears in the proof of Theorem 3.1. As a consequence, the Pareto front belongs to the surface determined by $Q = G^{-1}$.

Consider $s' = 2P_{\mathcal{A}^*}s - s$. It is straightforward to see that $\|s'\| = 1$ since it is the reflection of $s$ w.r.t. $\mathcal{A}^*$. To show 2), we will prove:

$$\langle s', x \rangle \geq |\langle s, x \rangle|, \quad \forall x \in \mathcal{A}. \tag{40}$$

We break the discussion into two cases.

**Case 1:** $\langle s, x \rangle \geq 0$. Since $x \in \mathcal{A} \subset \mathcal{A}^*$, we have

$$\langle s', x \rangle \geq \langle s, x \rangle = |\langle s, x \rangle| \tag{41}$$

by Lemma B.6.

**Case 2:** $\langle s, x \rangle < 0$. Consider $u = -s$ and denote $\mathcal{C} := \{a \mid \langle a, P_{\mathcal{A}^*}s \rangle \geq 0\}$. It is straightforward to see that $\mathcal{A} \subset \mathcal{C}$.

We will show $P_\mathcal{C} u = P_{\mathcal{A}^*}s - s$. In fact, for any $z \in \mathcal{C}$, we have

$$\begin{aligned} \|z - u\|^2 &= \|z - (P_{\mathcal{A}^*}s - s) + P_{\mathcal{A}^*}s\|^2 \\ &= \|z - (P_{\mathcal{A}^*}s - s)\|^2 + \|P_{\mathcal{A}^*}s\|^2 + 2\langle z - (P_{\mathcal{A}^*}s - s), P_{\mathcal{A}^*}s \rangle \\ &\geq \|P_{\mathcal{A}^*}s\|^2 + 2\langle z, P_{\mathcal{A}^*}s \rangle \\ &\geq \|P_{\mathcal{A}^*}s\|^2 \\ &= \|P_{\mathcal{A}^*}s - s - u\|^2. \end{aligned}$$

Therefore, we have

$$s' = 2P_{\mathcal{A}^*}s - s = 2P_\mathcal{C}u - u. \tag{42}$$

With another application of Lemma B.6, we have

$$\langle s', x \rangle \geq \langle u, x \rangle = \langle -s, x \rangle = |\langle s, x \rangle|. \tag{43}$$

This finishes the proof of the *if* part.

For the *only if* part, we consider the point $p_i$ which achieves maximum value along the $i$-axis. For a given $i$, it is straightforward to see that such $p_i$ is unique, and therefore is a PO of the feasible region.

Now, $p_i$ belongs to a surface determined by $Q'$, if and only if there exists a non-negative vector $v_i$, such that $Q'v_i = e_i$ (the normal vector at $p_i$ aligns with the $i$-th axis). In other words, $G'e_i \geq 0$ where $G' = (Q')^{-1}$. When C1 does not hold, there does not exist a $G^* \in \mathcal{G}$, such that $G^*e_i \geq 0$ for all $i \in [k]$. This implies that these $p_i$ must belong to different surfaces.

$\square$

## B.2 Proof of Lemma 3.5

*Proof of Lemma 3.5.* For the *if* part, we assume w.l.o.g. that $Q_\varnothing = G_\varnothing^{-1}$ is doubly non-negative. This essentially implies that $E_\varnothing$ is dominated by all other surfaces in $\mathcal{E}$, which further implies that $I_\varnothing$ dominates all other surfaces in $\mathcal{I}$. Therefore, the Pareto front must belong to $I_\varnothing$.

For the *only if* part, we study the Pareto front of $\mathcal{F}_{q-1}$ through the lens of $\mathcal{F}_1$. Specifically, a point $z$ is a PO of $\mathcal{F}_{q-1}$ if and only if $\mathcal{F}_{q-1}$ and the non-negative orthant of $z$ intersects only at $z$. Denote the corresponding point of $z$ on $\mathcal{F}_1$ as $z' = t - z$. An equivalent characterization is that $\mathcal{F}_1$ and the non-positive orthant of $z'$ intersects only at $z'$. We can therefore consider a special type of $z'$ whose coordinates are zero except for $i, j$-th entry. If $z'$ further lies on a surface $Q$ with $q_{ij} > 0$, then the corresponding $z$ will be a PO of $\mathcal{F}_{q-1}$.

When C2 does not hold, there exists a row of $Q_\varnothing$ which contains both positive and negative entries. We assume w.l.o.g. the first row of $Q_\varnothing$ satisfies this condition. By the above observation, we can find two PO of $\mathcal{F}_{q-1}$ that correspond to $Q_\varnothing$ and $Q_1$, respectively. This finishes the proof as desired. $\square$

## B.3 Proof of Theorem 3.8

*Proof of Theorem 3.8.* We only prove for the case of $q = 1$, and the proof for $q = k - 1$ can be done similarly. Suppose $P$ (whose coordinate is $v$) lies at the intersection of two surfaces defined by $Q$ and $Q'$. We denote the intersection of $Q$ and $Q'$ as $\mathcal{S}$, which is a non-linear manifold with dimension $(k - 2)$. Since $P$ is a relative interior point of the PF, there exists some $\varepsilon > 0$, such that any point in $S \cap B_\varepsilon(P)$ is a PO. We will show there exists some $P' \in S \cap B_\varepsilon(P)$ (whose coordinate is $v'$), such that the gradients at $P'$ w.r.t. the two surfaces disagree, i.e.,

$$Q\sqrt{v'} \neq Q'\sqrt{v'}. \tag{44}$$

To see why this is true, note that $Q'$ can be expressed as $Q' = DQD$, where $D$ is a diagonal matrix whose diagonal entries are either $1$ or $-1$. As a consequence, the set

$$\{v \mid Qv = Q'v\} \tag{45}$$

is a subspace of $\mathbb{R}^k$ whose dimension is at most $(k - 2)$, so it cannot fully contain a non-linear manifold whose dimension is $(k - 2)$. This finishes the proof as desired. $\square$

## B.4 Proof of Theorem 3.8 without assumptions ($k = 3$)

Here we provide a proof for Theorem 3.8 under $k = 3$, without relying on Assumptions 3.6 and 3.7.

*Proof.* We prove the necessity of C1 and C2 separately.

**C1 is necessary.** Suppose C1 is not true. We assume w.l.o.g. that $\langle \hat{y}_1, \hat{y}_2 \rangle < 0, \langle \hat{y}_1, \hat{y}_3 \rangle > 0, \langle \hat{y}_2, \hat{y}_3 \rangle > 0$. Now we can write

$$Q = \begin{bmatrix} q_{11} & q_{12} & -q_{13} \\ q_{21} & q_{22} & -q_{23} \\ -q_{31} & -q_{32} & q_{33} \end{bmatrix}, \tag{46}$$

where $q_{ij} > 0$ for all pairs of $(i, j)$. We also have

$$q_{11}q_{23} > q_{12}q_{13}, \; q_{22}q_{13} > q_{12}q_{23}, \; q_{33}q_{12} > q_{13}q_{23}. \tag{47}$$

The boundary of the feasible region is formed by $E_\varnothing, E_1, E_2$. Our goal is to find a point $I \in E_\varnothing \cap E_1$, such that 1) $I$ is a PO of the feasible region; 2) the gradients at $I$ w.r.t. $E_\varnothing$ and $E_1$ disagree. We write the coordinate of $I$ as $(x, y = q_{13}^2/s, z = q_{12}^2/s)^\top$, where

$$q_{11}x + \frac{q_{22}q_{13}^2 + q_{33}q_{12}^2 - 2q_{12}q_{23}q_{31}}{s} = 1, \tag{48}$$

and $s < q_{11}q_{23}^2 + q_{22}q_{13}^2 + q_{33}q_{12}^2 - 2q_{12}q_{23}q_{31}$.

It is straightforward to see that 2) can be satisfied as long as $x \neq 0$. Therefore, we will focus on the realization of 1). This further requires two conditions: i) $I$ is a PO w.r.t. $E_\varnothing$ and $E_1$, meaning that the normal vectors are non-negative; ii) $I$ is a PO w.r.t. $E_2$, meaning that the non-negative orthant at $I$ does not intersect with $E_2$.

Some simple calculations yield that i) is equivalent to

$$q_{22}\sqrt{y} \geq q_{23}\sqrt{z} + q_{12}\sqrt{x} \quad \text{and} \quad q_{33}\sqrt{z} \geq q_{23}\sqrt{y} + q_{13}\sqrt{x}. \tag{49}$$

Setting $x = 0$, we have

$$q_{22}\sqrt{y} > q_{23}\sqrt{z} \quad \text{and} \quad q_{33}\sqrt{z} > q_{23}\sqrt{y} \tag{50}$$

by Equation (47). Therefore, it suffices to show the non-negative orthant at $(0, q_{13}^2/s, q_{12}^2/s)^\top$ ($s = q_{22}q_{13}^2 + q_{33}q_{12}^2 - 2q_{12}q_{23}q_{31}$) does not intersect with $E_2$, and by continuity we can find some $x_0 > 0$ such that i) and ii) hold simultaneously.

To prove the above claim, suppose $a \geq 0, b \geq q_{13}^2/s, c \geq q_{12^2}/s$. We will show $(a, b, c)^\top$ must lie above the surface defined by $E_2$, i.e.,

$$f(a, b, c) = q_{11}a + q_{22}b + q_{33}c - 2q_{12}\sqrt{ab} - 2q_{13}\sqrt{ac} + 2q_{23}\sqrt{bc} > 1. \tag{51}$$

Let $a^* = \frac{q_{12}\sqrt{b}+q_{13}\sqrt{c}}{q_{11}}$. We have

$$
\begin{aligned}
f(a,b,c) &\geq f(a^*,b,c) \\
&= \left(q_{22} - \frac{q_{12}^2}{q_{11}}\right)b + \left(q_{33} - \frac{q_{13}^2}{q_{11}}\right)c + 2\left(\frac{q_{11}q_{23} - q_{12}q_{13}}{q_{11}}\right)\sqrt{bc} \\
&\geq \frac{q_{11}(q_{22}q_{13}^2 + q_{33}q_{12}^2 + 2q_{12}q_{23}q_{31}) - 4q_{12}^2 q_{13}^2}{sq_{11}} \\
&> \frac{q_{11}(q_{22}q_{13}^2 + q_{33}q_{12}^2 - 2q_{12}q_{23}q_{31})}{sq_{11}} \\
&= 1.
\end{aligned}
$$

As a consequence, by choosing a small $x_0$, we can guarantee both 1) and 2). $I$ is therefore a PO that cannot be achieved via scalarization.

C2 **is necessary.** Suppose C2 is not true. We can write

$$
Q = \begin{bmatrix} q_{11} & -q_{12} & -q_{13} \\ -q_{21} & q_{22} & -q_{23} \\ -q_{31} & -q_{32} & q_{33} \end{bmatrix}, \tag{52}
$$

where $q_{ij} > 0$ for all pairs of $(i,j)$. The inner boundary of $\mathcal{F}_1$ is formed by $E_1, E_2, E_3$, and the coordinates of their intersection $I$ is given by $v = (q_{23}^2/r, q_{13^2}/r, q_{12}^2/r)^\top$, where $r = q_{11}q_{23}^2 + q_{22}q_{13}^2 + q_{33}q_{12}^2 + 2q_{12}q_{23}q_{31}$. Our goal is to show the corresponding $I'$ on $\mathcal{F}_2$ (i.e., $t - v$) is a PO which cannot be achieved by scalarization.

To see why this is true, first note that the gradients at $I$ w.r.t. the three surfaces disagree, since $q_{ij} > 0$ for all pairs of $(i,j)$. Now, to demonstrate $I'$ is a PO of $\mathcal{F}_2$, it suffices to show the non-positive orthant of $I$ intersects with $\mathcal{F}_1$ only at $I$. By symmetry, it suffices to show that $I$ does not dominate any point on $E_1$.

To prove the above claim, suppose $a \leq q_{23}^2/r, b \leq q_{13}^2/r, c \leq q_{12}^2/r$. We will show, $J = (a,b,c)^\top$ must lie below the surface defined by $E_1$ unless $J$ equals $I$. In fact, let

$$
g(a,b,c) = q_{11}a + q_{22}b + q_{33}c + 2q_{12}\sqrt{ab} + 2q_{13}\sqrt{ac} - 2q_{23}\sqrt{bc}. \tag{53}
$$

We have

$$
\begin{aligned}
g(a,b,c) &\leq g(q_{23}^2/r, b, c) \\
&\leq g(q_{23}^2/r, q_{13}^2/r, c) \\
&\leq g(q_{23}^2/r, q_{13}^2/r, q_{12}^2/r) \\
&= 1,
\end{aligned}
$$

where the equalities hold iff $I = J$. This finishes the proof as desired. $\qquad\square$

## B.5 Proof of Theorem 3.10

We first present several useful lemmas regarding non-negative matrices.

**Lemma B.7.** *Let $A$ be a non-negative, irreducible and real symmetric matrix, then there only exists one eigenvector $v$ of $A$ (up to scaling), such that $v > 0$.*

*Proof.* Since $A$ is non-negative and irreducible, by Perron-Frobenius theorem (Shaked-Monderer and Berman, 2021), the eigenvector associated with the largest eigenvalue $\rho(A)$ (a.k.a. *Perron root*) can be made positive, and is unique up to scaling. We denote it as $v$.

We will show the eigenvectors associated with other eigenvalues cannot be positive. In fact, assume $u > 0$ is an eigenvector associated with $\lambda < \rho(A)$. Since $A$ is symmetric, we have

$$
\rho(A)v^\top u = v^\top A^\top u = v^\top A u = \lambda v^\top u. \tag{54}
$$

This is a contradiction since $v^\top u > 0$ and $\rho(A) > \lambda$. $\qquad\square$

**Lemma B.8.** *Let $A$ be a non-negative and real symmetric matrix. Suppose $v$ is an eigenvector of $A$ such that $v > 0$, then $v$ corresponds to the largest eigenvalue of $A$.*

*Proof.* Consider the normal form (Varga, 1999) of $A$:

$$PAP^\top = \begin{bmatrix} B_{11} & B_{12} & \cdots & B_{1h} \\ 0 & B_{22} & \cdots & B_{2h} \\ \vdots & \vdots & \ddots & \vdots \\ 0 & 0 & \cdots & B_{hh} \end{bmatrix} \triangleq B, \tag{55}$$

where $P$ is a permutation matrix and $B_{ii}$ ($i \in [h]$) are irreducible. It is straightforward to see that $B$ is non-negative. Furthermore, since a permutation matrix is an orthogonal matrix, $B$ is also symmetric (implying that $B_{ij} = 0$ for $i \neq j$) and has the same spectrum as $A$.

Now suppose $Av = \lambda v$ for some $v > 0$. Since $B = PAP^\top$, we have $B\psi = \lambda\psi$ with $\psi = Pv > 0$. Denote

$$\psi = \begin{pmatrix} \psi_1 \\ \psi_2 \\ \vdots \\ \psi_n \end{pmatrix}, \tag{56}$$

where each $\psi_i$ has the same dimension as $B_{ii}$. Now we have

$$B_{ii}\psi_i = \lambda\psi_i, \quad \forall i \in [h]. \tag{57}$$

Since $B_{ii}$ is non-negative, irreducible and real symmetric, by Lemma B.7, $\psi_i$ corresponds to the largest eigenvalue of $B_{ii}$. Note $B$ is a block diagonal matrix, so the spectrum of $B$ is essentially the union of the spectrum of $B_{ii}$. As a consequence,

$$\lambda_{\max}(A) = \lambda_{\max}(B) = \max_{i \in [h]} \lambda_{\max}(B_{ii}) = \lambda. \tag{58}$$

$\square$

*Proof of Theorem 3.10.* We prove the sufficiency of `C1` and `C2` separately.

`C1` **is sufficient.** We have demonstrated in Lemma 3.4 that the Pareto front belongs to a single surface $E$, whose gram matrix is doubly non-negative. Let $p = (a_1, \cdots, a_k)^\top$ be a PO of $E$, and denote $t = \sqrt{p}$. We will show that the tangent plane at this point lies above the feasible region, implying that it can be achieved by scalarization.

Concretely, the tangent plane is given by

$$\sum_{i \in [k]} \frac{\sum_{j \in [k]} q_{ij}\sqrt{a_j}}{\sqrt{a_i}} v_i = 1. \tag{59}$$

To show it lies above the feasible region, it suffices to prove that it lies above every surface in $\mathcal{E}$, i.e.,

$$\sum_{i \in [k]} \frac{\sum_{j \in [k]} q_{ij}\sqrt{a_j}}{\sqrt{a_i}} v_i \leq \sum_{i \in [k]} q_{ii}^{i_1 \cdots i_l} v_i + 2 \sum_{1 \leq i < j \leq k} q_{ij}^{i_1 \cdots i_l} \sqrt{v_i v_j}, \tag{60}$$

where $q_{ij}^{i_1 \cdots i_l}$ represents the $(i, j)$-th entry of $Q_{i_1 \cdots i_l}$. Note Equation (60) can be treated as a quadratic form w.r.t. $\sqrt{v}$, so it suffices to prove the corresponding matrix is PSD. Writing compactly, it suffices to show

$$Q_{i_1 \cdots i_l} - \mathrm{diag}\{Qt \oslash t\} = D_{i_1 \cdots i_l}(Q - \mathrm{diag}\{Qt \oslash t\})D_{i_1 \cdots i_l} \tag{61}$$

is PSD. As a consequence, we only need to show the positive semi-definiteness of the following matrix

$$T := Q - \mathrm{diag}\{Qt \oslash t\}. \tag{62}$$

Since $p$ is a PO of $E$, we have $Qt \geq 0$. Let $w = Qt$, we have

$$T = G^{-1} - \mathrm{diag}\{w \oslash Gw\}. \tag{63}$$

Assume w.l.o.g. that $w > 0$ (otherwise we simply remove the corresponding rows and columns in $G^{-1}$). Denote

$$R = \sqrt{\text{diag}\{w \oslash Gw\}} \quad \text{and} \quad \phi = \sqrt{w \odot Gw} > 0. \tag{64}$$

We have

$$
\begin{aligned}
RGR\phi &= RG\sqrt{\text{diag}\{w \oslash Gw\}}\sqrt{w \odot Gw} \\
&= RGw \\
&= \sqrt{\text{diag}\{w \oslash Gw\}}Gw \\
&= \sqrt{w \odot Gw} \\
&= \phi.
\end{aligned}
$$

By Lemma B.8, we have $\rho(RGR) = 1$. Since $RGR$ is positive definite, we have

$$I \succeq RGR \succ 0. \tag{65}$$

This further implies that $R^{-2} \succeq G$, and

$$T = G^{-1} - R^2 \succeq 0. \tag{66}$$

**C2 is sufficient.** We have demonstrated in Lemma 3.5 that the Pareto front belongs to a single surface $I_\varnothing$. Here we will further show $I_\varnothing$ is concave, and apply the supporting hyperplane theorem (Boyd et al., 2004, Section 2.5.2) to conclude the proof. In fact, $E_\varnothing$ is parametrized by

$$\sum_{i \in [k]} q_{ii} v_i + 2 \sum_{1 \leq i < j \leq k} q_{ij} \sqrt{v_i v_j} = 1. \tag{67}$$

Since the first term is a summation of linear functions, $f(x, y) = \sqrt{xy}$ is a concave function and that $q_{ij} \geq 0$, we conclude that $E_\varnothing$ is convex, and $I_\varnothing$ is concave. This finishes the proof as desired. $\qquad\square$

# C  A polynomial algorithm for checking C1

We explain how Algorithm 1 works.

Essentially, the goal is to find a set of flipping signs $\{s_i\}_{i \in [k]} \in \{+, -\}$ that ensure positive correlation between all pairs in $\{s_i \hat{y}_i\}_{i \in [k]}$ (except for those pairs that are not correlated). We state an observation which is critical for our algorithm—given a pair $\hat{y}_i$ and $\hat{y}_j$, whether their signs are the same or opposite can be determined by whether $\hat{y}_i$ and $\hat{y}_j$ is positively or negatively correlated. Thus, given a determined $s_i$, the signs of its subsequent predictors $s_j$, $j > i$ can be uniquely determined by evaluating $\langle \hat{y}_i, \hat{y}_j \rangle$. This allows us to determine the signs for the predictors $s_i$ in ascending order, and check for potential conflicts in the mean time.

We state one loop invariant in our algorithm—when entering iteration $i$ in the outer loop, $s_i$ has been determined by its *preceding* predictors and no conflict has occurred, or it has remained undetermined (i.e., None) all the way because it is not correlated with any preceding predictors. At this point, what is left to be done is to examine the *subsequent* predictors. If $s_i$ is undetermined (lines 2-7), we attempt to use its subsequent predictors to determine it (line 5), and break out of the loop once it gets determined. Now that we make sure that $s_i$ is determined after line 9, we can proceed to check the remaining subsequent predictors, to either determine their signs (lines 12-13) or check for potential conflicts (lines 14-15).

We comment that caution needs to be taken for the pairs that are not correlated (i.e., $\langle \hat{y}_i, \hat{y}_j \rangle = 0$). In this case, one cannot determine the sign of $s_j$ by the relationship between these two predictors; basically, the sign has no influence on their correlation. Thus, the predictor $\hat{y}_j$ retains the flexibility to be determined by other predictors other than $\hat{y}_i$. Additionally, it is possible that at the end of line 9 $s_j$ is still undetermined; this only occurs when $\hat{y}_i$ is not correlated with any other optimal predictor. In this case, $t$ will take the value of $k + 1$ and the following loop (lines 10-15) will not be executed.

# D  Additional experimental details and results

## D.1  Additional experimental details

**Dataset.** The SARCOS dataset (Vijayakumar and Schaal, 2000) consists of the configurations (position, velocity, acceleration) of the robotic arms, and the problem is to predict the torque of the

**Algorithm 1:** An $O(k^2)$ algorithm of checking `C1`

---

**Input:** Optimal predictors $\hat{y}_1, \hat{y}_2, \ldots, \hat{y}_k$
**Output:** `True` if `C1` is met, and `False` otherwise
**Initialize:** $s_1 = 1$, $s_2 = \cdots = s_k = $ `None`
**Define:** $\mathrm{sgn}(x) = \begin{cases} 1, & \text{if } x > 0 \\ -1, & \text{if } x < 0 \\ \texttt{None}, & \text{if } x = 0 \end{cases}$

**1**   **for** $i = 1$ **to** $k - 1$ **do**
     ▷ Check whether $s_i$ has been determined by its preceding optimal predictors
     $\{\hat{y}_1, \ldots, \hat{y}_{i-1}\}$; if not, determine it using the subsequent ones.
**2**     **if** $s_i = $ `None` **then**
**3**        **for** $j = i + 1$ **to** $k$ **do**
**4**           **if** $s_j \neq $ `None` ***and*** $\mathrm{sgn}(\langle \hat{y}_i, \hat{y}_j \rangle) \neq $ `None` **then**
**5**              $s_i \leftarrow s_j \cdot \mathrm{sgn}(\langle \hat{y}_i, \hat{y}_j \rangle)$
**6**              **break**
**7**        $t \leftarrow j + 1$
**8**     **else**
**9**        $t \leftarrow i + 1$
     ▷ Now that $s_i$ has been determined, we proceed to check the remaining optimal
     predictors to either determine the sign for them or check for conflicts.
**10**    **for** $j = t$ **to** $k$ **do**
**11**       **if** $\mathrm{sgn}(\langle \hat{y}_i, \hat{y}_j \rangle) \neq $ `None` **then**
**12**          **if** $s_j = $ `None` **then**
**13**             $s_j \leftarrow s_i \cdot \mathrm{sgn}(\langle \hat{y}_i, \hat{y}_j \rangle)$
**14**          **else if** $s_i \cdot s_j \cdot \mathrm{sgn}(\langle \hat{y}_i, \hat{y}_j \rangle) = -1$ **then**
**15**             **return** `False`

**16** **return** `True`

---

respective arm given its triplet configuration. Since our study concentrates on the training procedure and does not concern generalization, we conduct the experiments on the training set alone, where the size of the training set of the SARCOS dataset (Vijayakumar and Schaal, 2000) is $40,000$. Regression tasks normally benefit from standardization as a pro-processing step (Hastie et al., 2009), so we standardize the dataset to zero mean and unit variance.

**Tasks.** Following our theoretical analysis in Section 3, we study the scenario where $q = 1$. In this case, it is straightforward to see that the minimal $k$ that may lead to a violation of `C1` is 3. To show the violation of `C1` leading to the failure of scalarization exploring the Pareto Front, we constrain our study to $k \geq 3$; furthermore, for ease of visualization, we focus on $k = 3$. In our experiments, we take the arms 3, 4, and 5 for which do not satisfy `C1`.

**Network.** We use a two-layer linear network for the reason explained in Section 2. We merge the bias term $b$ into the weight matrix $W$ by adding an additional all-one column to the input $X$. The input dimension of the network is 22 (seven (position, velocity, acceleration) triplets plus the dummy feature induced by $b$), the hidden size is $q = 1$, and the output size is 1 (i.e., the predicted torque). The same network architecture is used for both experiments in scalarization and that in SMTOs.

**Implementation of linear scalarization.** For linear scalarization, we uniformly sample $100,000$ sets of convex coefficients from a three-dimensional simplex (i.e., a tetrahedron). Concretely, we first sample $m_1, m_2 \overset{i.i.d.}{\sim} U(0, 1)$, and then craft $\boldsymbol{\lambda} = (\lambda_1, \lambda_2, \lambda_3) = (\min(m_1, m_2), \max(m_1, m_2) - \min(m_1, m_2), 1 - \max(m_1, m_2) + \min(m_1, m_2))$ as the weights. For a given set of convex coefficients, we calculate the optimal value directly based on the analysis in Section 2 instead of performing actual training. More specifically, the optimum of the scalarization objective is given by $\hat{Y}_{q,\Lambda}\Lambda := \sum_{i=1}^{q} \sigma_i u_i v_i^\top$, from which we can compute the corresponding MSE for each task.

**SMTOs.** We consider two state-of-the-art SMTOs, MGDA (Désidéri, 2012) and MGDA-UB (Sener and Koltun, 2018). MGDA is an optimization algorithm specifically devised for handling multiple objectives concurrently. It utilizes a composite gradient direction to facilitate simultaneous progress across all objectives. MGDA-UB is an efficient variant of MGDA that focuses on maximizing the minimum improvement among all objectives. It is a gradient-based multi-objective optimization algorithm aimed at achieving balanced optimization outcomes across all objectives.

**Implementation of SMTOs.** Our code is based on the released implementation[5] of MGDA-UB, which also includes the code for MGDA. We apply their code on the SARCOS dataset. For both methods, we use vanilla gradient descent with a learning rate of $0.5$ for $100$ epochs, following the default choice in the released implementation. We comment that early stopping can also be adopted, i.e., terminate once the minimum norm of the convex hull of gradients is smaller than a threshold, for which we set to be $10^{-3}$.

### D.2 Additional experiments on random initialization

To eliminate the influence of random initialization, we perform 300 trials for each algorithm using different random seeds. We filter out solutions whose maximum MSE is larger than 1 for clearer presentation, which leads to 198 solutions for MGDA and 240 solutions for MGDA-UB. We plot all these solutions in Figure 4. We see that MGDA and MGDA-UB are consistently capable of finding balanced solutions regardless of the random seed.

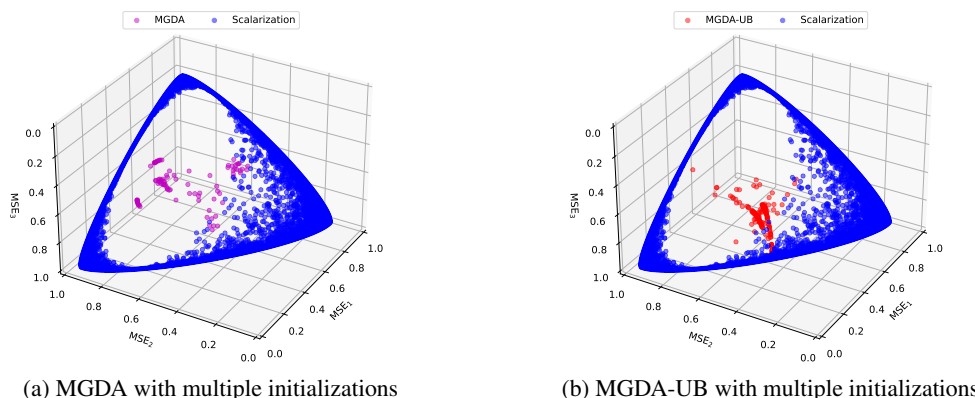

(a) MGDA with multiple initializations      (b) MGDA-UB with multiple initializations

Figure 4: More solutions of MGDA and MGDA-UB obtained by varying the initializations. Both algorithms tend to find solutions located at the interior of the Pareto front.

### D.3 Additional experiments on randomization

To verify the effectiveness of randomization as discussed in Section 4.2, we plot the region achievable by randomized linear scalarization. Specifically, according to Equation (13), the region can be expressed as the collection of the convex combination of two points from the feasible region. To this end, we randomly sample 100,000 weight pairs (the same number as in Figure 3). For each weight pair $(w_1, w_2)$, we uniformly draw $t \sim U(0, 1)$ and get two corresponding optimal networks $f_1$ and $f_2$ by SVD. For each sample, with probability $t$, the model uses $f_1$, otherwise $f_2$, to calculate the MSE. The final result is demonstrated in Figure 5. It is straightforward to see that randomization allows scalarization to trace out the PF since there is no hole within the blue region, thus validating our theoretical analysis. We additionally comment that randomization convexifies the feasible region, as such, the solutions found by MGDA and MGDA-UB are dominated.

---

[5]`https://github.com/isl-org/MultiObjectiveOptimization`

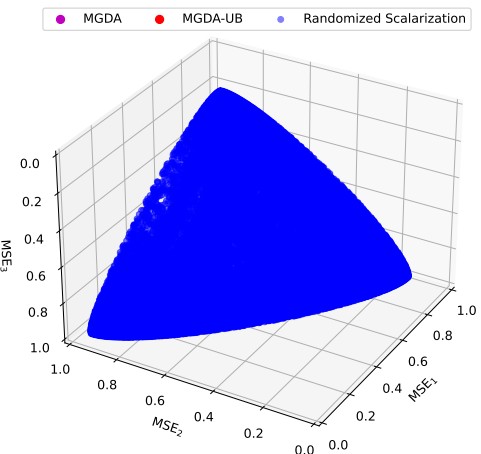

Figure 5: The region achievable by randomized linear scalarization

