# OpenReview forum: "Revisiting Scalarization in Multi-Task Learning: A Theoretical Perspective"
_NeurIPS.cc/2023/Conference — NeurIPS 2023 poster_

### Official Review · Reviewer_aGcP · 2023-06-26

**Soundness:** 3 good
**Presentation:** 2 fair
**Contribution:** 3 good
**Rating:** 7
**Confidence:** 4

**Summary:**

In recent years, there has been a surge in papers suggesting Specialized Multi-Task Optimizers (SMTOs). These papers show the empirical advantage of using SMTOs compared to linear scalarization (LS). However, recently, there have been several papers that suggested that LS with proper tunning can match SMTOs performance. The paper analyzes scalarization from a theoretical perspective, and studies whether scalarization is capable of fully exploring the Pareto front in linear MTL. It reveals some inherent limitations of LS for MTL optimization.

**Strengths:**

1. The paper addresses an important open question in MTL optimization.
1. The paper is (mostly) well-written and well-structured.
1. The paper provides an in-depth theoretical analysis with implications for both MTL researchers and practitioners.


**Weaknesses:**

1. A primary argument is that LS cannot reach a point in the intersection of surfaces. However, it is not clear from Section 3.2 why this is the case. Please elaborate on this point and revise the manuscript accordingly.
1. Please discuss the main results and novelty w.r.t the known result ([1]), which states that LS cannot reach non-convex parts of the Pareto front.
1. It would be beneficial to revise Section 3, by either breaking it into two sections or summarizing the main results at the beginning of the Section.
1. Please add an experiment with the proposed randomization approach to empirically verify that it can explore the entire Pareto front.
1. Missing citations in related work:
    1. SMTOs:
        - Multi-Task Learning as a Bargaining Game, ICML 2022.
        - Towards Impartial Multi-task Learning, ICLR 2021.
        - Gradient Vaccine: Investigating and Improving Multi-task Optimization in Massively Multilingual Models, ICLR 2021.
    1. Exploration of the Pareto front: There’s a line of work for methods trying to learn the entire Pareto front, so it is important to mention it. The two pioneering papers for Pareto front learning are:
        - Learning the Pareto Front with Hypernetworks, ICLR 2021.
        - Controllable Pareto Multi-Task Learning, 2021.

[1] S. Boyd, S. P. Boyd, and L. Vandenberghe. Convex optimization. 2004.


**Questions:**

1. How realistic is the case $q<k$ for real-world applications?

**Limitations:**

Addressed.

---

> ### Author Rebuttal · Authors · 2023-08-10
>
> We would like to thank Reviewer aGcP for their constructive feedback. We are grateful that the reviewer appreciates the significance of the problem we tackled as well as the theoretical contributions. We hope to address all points in the review below, following the order they were made.
>
> **Why scalarization cannot explore intersection points.** Please refer to point 3 of the general response, in which we provide detailed reasoning steps and explanations. We will elaborate on this point in the revision.
>
> **Comparison with [1].** We have carefully checked the reference by searching the keyword 'scalarization', but were unable to find the result that you mentioned. Could you kindly refer to the relevant content in the book? As far as we know, [1] only states that, when the objective functions are convex, the Pareto front can be fully explored by LS. Figure 4.9 in [1] provides an example of PO ($f_0(x_3)$) that cannot be achieved by LS, but it is unclear how it corresponds to the 'non-convex part' of the PF, and how $f_0(x_1)$ and $f_0(x_2)$ corresponds to 'convex parts' of the PF. We believe that these terms (convex or non-convex part of a set) are not well-defined, as one can only talk about the convexity or non-convexity of the PF as a whole.
>
> In our work, we note that the PF is only a subset of the feasible region. Proving the PF is convex does not directly yield the result of full-exploration. As a consequence, we approach this problem from a different perspective. After revealing the multi-surface structure of the feasible region, we examine when the PF lies on a single surface instead of checking when it is convex. This allows us to derive a both necessary and sufficient condition for full-exploration.
>
> **Structure of Section 3.** Thanks for the suggestion! We agree that the current Section 3 is too long, and will break it into two sections to increase the readability.
>
> **Experiments on randomization.** We conduct experiments based on the equation derived before Line 300. Specifically, the region achievable by randomization can be expressed as the collection of the convex combination of two points from the feasible region. To this end, we randomly sample 100000 weight pairs (the same number as in the original experiment). For each weight pair $(w_1,w_2)$, we uniformly draw $t \sim U(0,1)$, and get two corresponding optimal networks $f_1$ and $f_2$ by SVD. For each sample, with probability $t$, the model uses $f_1$, otherwise $f_2$, to calculate the MSE. Finally, we plot the set of all MSEs in Figure B (see the attached PDF in the general response). It is straightforward to see that randomization allows scalarization to trace out the PF since there is no hole within the blue region, thus validating our theoretical analysis. We additionally comment that randomization convexifies the feasible region, as such, the solutions found by MGDA and MGDA-UB are dominated.
>
>
> **Missing references.** We thank the reviewer for pointing out the references, and will for sure incorporate them in the revision.
>
> **The $q<k$ assumption.** Please refer to the first point of the general response.
>
>
> **References**
>
> [1] Boyd, Stephen P., and Lieven Vandenberghe. Convex optimization. Cambridge university press, 2004.

---

> > ### Comment · Reviewer_aGcP · 2023-08-15
> >
> > Thanks for the response and for providing additional results.
> >
> > -  **Comparison with [1]:** I meant the results in Ch 4.7 of [1], specifically, please see Ch. 4.7.4. Note that Linear Scalarization is referred to as Scalarization.
> > - **Point raised by Reviewer nBcV w.r.t Xin et al. (2022):** I agree with the point raised by Reviewer nBcV regarding the results w.r.t Xin et al. (2022). The theoretical results here are limited in terms of applicability to real-world scenarios and modern, over-parametrized MTL networks. This does not diminish or damage the value and importance of the analysis in the paper, but the limitations in immediate empirical implication should be made clear.
> >
> >
> > [1] S. Boyd, S. P. Boyd, and L. Vandenberghe. Convex optimization. 2004.

---

> > > ### Author Response · Authors · 2023-08-15
> > >
> > > Thanks for your response.
> > >
> > > Regarding the first point, we believe the results in Ch. 4.7.4 have been well summarized in Xin et al. (2022) (see Theorem (Informal) in P3). Specifically, 1) any solution to the scalarization objective with positive coefficients is Pareto optimal; 2) when the objective functions are convex, the Pareto front can be fully explored by linear scalarization. In contrast, our theoretical results differ significantly from prior works because 1) they are established in the non-convex setting, which demands fundamentally different techniques compared to those used in convex analysis; 2) the conditions for full exploration that we have uncovered are both necessary and sufficient, which helps us understand the weakness of scalarization; 3) finally, we build up a theory (the multi-surface structure and the phenomenon of gradient disagreement) to explain why scalarization fails. This significantly advances previous studies by offering new insights into the failure modes of scalarization, going beyond mere hypothetical examples (e.g., Figure 4.9 in the textbook).
> > >
> > > We agree with the reviewer that it is important to contrast our results with prior art in order to showcase their significance and novelty. We will make sure that the above points are adequately discussed in the revision.
> > >
> > > Regarding the second point, we would like to clarify that we do not attempt to completely refute the claims and results in Xin et al. (2022), and we understand the difference in the settings makes the results in these two papers not directly comparable. Instead, below are our standing points:
> > > - We feel there are some reasoning gaps in Xin et al. (2022) that we would like to point out and clarify. Specifically, as shown in our paper, the positive results in the convex setting do not transfer. Therefore, the theory they developed cannot be used to support their experiments, which are performed with deep neural networks. By pointing this out, we call for a need from the research community to develop a new theory to explain the empirical success of linear scalarization.
> > > - We don't think one single SMTO algorithm can fix the issue of linear scalarization, and it is not our goal to advocate for a particular algorithm like MGDA. Instead, we hope to refute the claim that linear scalarization is sufficient for MTL, and bolster research in the development of SMTO algorithms. One of our central goals is to promote a healthier and more balanced algorithmic developement in the field of MTL.
> > > - We view our work as a complement to Xin et al. (2022) (as well as a few other works). Putting together, they provide a more comprehensive view on the strengths and weakness of linear scalarization and SMTO.
> > >
> > > We thank the reviewer for emphasizing this point. We will make sure to discuss the above points as well as the limitation of our empirical evaluations in the revision.
> > >
> > > Again, we would like to thank the reviewer for taking the time to review our paper and joining the discussion. We will be happy to answer any further questions.

---

### Official Review · Reviewer_CFe6 · 2023-07-03

**Soundness:** 3 good
**Presentation:** 3 good
**Contribution:** 3 good
**Rating:** 5
**Confidence:** 2

**Summary:**

This paper is related to multi-objective optimization area.
They post a research question: if linearly weighting multiple objectives can fully explore the Pareto front?
Through theoretical analysis and a simple experiment, the answer is negative.
Hence, this might prove that multiple gradient descent algorithm is inherently better than linear objective weighting.

**Strengths:**

Recently, many report that linear weighting is no worse and even better than gradient balancing algorithms like multiple gradient descent algorithm (MGDA).
However, these findings are empirical,
This paper targets on an important question: if multiple gradient descent algorithm is inherently better than simple linear weighting when conflict gradients are presenting.
Through theoretical analysis and a simple experiment, the answer is yes, which lay a good foundation for this area.


**Weaknesses:**

The MTL model is this paper is very simple. I am not sure if this could reflect the real problem in the industry system.
This model only have one layer of shared layer and one layer of task-head. And this is a linear neural network. (I am not sure if pure linear can be called neural network).
So, seems this MTL model is a convex optimization.
However, in real industry systems, there are many layers (maybe millions of parameters) with non-linear transformation and hence a non-convex optimization.



**Questions:**

1. if MTL has different objectives like one regression task and one classification task, will your conclusion hold?

2. If MTL has non-linear transformation like Sigmoid, will your conclusion hold?

3. Do we have practical usage of randomization for linear scalarization?

**Limitations:**

No negative social impact

---

> ### Author Rebuttal · Authors · 2023-08-10
>
> We would like to thank Reviewer CFe6 for taking the time to review our paper. We appreciate that they found the problem of our study important. Below we attempt to address the reviewer’s concerns, following the order they were made.
>
> **Weaknesses.** Please refer to point 2 of the general response. We emphasize that even for two-layer linear networks, the loss function is *not* convex w.r.t. the model parameters, and that both the techniques and results in this paper are novel (to the best of our knowledge) and greatly advance those in the literature of convex analysis.
>
> **Question 1 and 2.** The honest answer to the first two questions is 'we don't know'. We understand that a more sophisticated setting or model architecture will be more appealing to the reviewer. However, we would like to share a few points in this aspect:
>
> - While a more complicated setting or model generally necessitates more advanced techniques, it is not the case conversely. For instance, ridge regression is probably one of the simplest models in machine learning, yet it is still being extensively studied in the literature [1,2,3]. In analogy, while we focus on linear MTL for regression, we are targeting an important question in MTL from an unique perspective, and the techniques that we have developed in this paper are highly non-trivial (e.g., the innovative application of Perron-Frobenius theorem, bridging full-exploration with doubly non-negative matrices, etc). We therefore believe the theoretical contribution of this paper is significant, and shouldn't be dimmed by the setting.
>
> - We carefully compile a list of implications derived through our theoretical analysis. We believe that sharing such insights with the broader community is more important than proving similar results in variations of the original setting.
>
> - Building up a strong and robust theory requires collaborative effort from the research community. As stated at the end of our paper, we hope this work can initiate a line of work that provides theoretical justifications of the usage of specific algorithms in MTL. We therefore believe that follow-up works will answer the reviewer's questions.
>
> We sincerely hope the reviewer can reconsider their evaluation of the paper based on the above response.
>
> **Question 3.** We are not aware of existing literature on using randomization in linear scalarization. But the caveat here is that randomization helps to enlarge the feasible regions beyond the ones that could be achieved by any deterministic models. Specifically, the new PF is going to be the convex hull of the original one, and this implies that 1) it can lead to better solution points (in the Pareto sense) overall; 2) it guarantees that the new PF can be fully explored by linear scalarization according to standard results in convex analysis (e.g., [4]). In essence, randomization increases the representation power of the original neural network.
>
> We additionally comment that randomization is a powerful tool that has wide applicability in machine learning. For instance, it has achieved great success in online learning, specifically in the Follow-the-Perturbed-Leader algorithm (see [5]). It also appears in the literature of fairness to achieve the optimal regressor or classifier (see [6]). In our work, we further demonstrate that it can be applied to MTL, specifically, to convexify the feasible region and allow linear scalarization to fully explore the PF.
>
>
> **References**
>
> [1] Richards, Dominic, Jaouad Mourtada, and Lorenzo Rosasco. "Asymptotics of ridge (less) regression under general source condition." International Conference on Artificial Intelligence and Statistics. PMLR, 2021.
>
> [2] Cheng, Chen, and Andrea Montanari. "Dimension free ridge regression." arXiv preprint arXiv:2210.08571 (2022).
>
> [3] Tsigler, Alexander, and Peter L. Bartlett. "Benign overfitting in ridge regression." J. Mach. Learn. Res. 24 (2023): 123-1.
>
> [4] Boyd, Stephen P., and Lieven Vandenberghe. Convex optimization. Cambridge university press, 2004.
>
> [5] https://users.soe.ucsc.edu/~sesh/Teaching/2020/CSE290A/Slides/Lecture18.pdf
>
> [6] Zhao, Han, and Geoffrey J. Gordon. "Inherent tradeoffs in learning fair representations." The Journal of Machine Learning Research 23.1 (2022): 2527-2552.

---

> > ### Comment · Reviewer_CFe6 · 2023-08-17
> > **reply to rebuttal**
> >
> > Thanks for the reply.
> >
> > I agree, although the network is simple in this paper, this is still an important step towards explaining the advantage of SMTO over linear weighting

---

> > ### Comment · Reviewer_CFe6 · 2023-08-17
> >
> > So I raised the rating from 4 to 5

---

> > > ### Author Response · Authors · 2023-08-17
> > >
> > > Thank you for taking the time to review our paper!

---

### Official Review · Reviewer_2hHW · 2023-07-06

**Soundness:** 3 good
**Presentation:** 3 good
**Contribution:** 3 good
**Rating:** 6
**Confidence:** 3

**Summary:**

This paper revisits linear scalarization from a theoretical perspective. The authors study multi-task learning with a two-layer linear network and reveal a multi-surface structure of the feasible region. They show the necessary and sufficient conditions for full exploration in the under-parameterized regime. Their theoretical results imply that linear scalarization has fundamental limitations and that scalarization tends to overfit a small fraction of tasks. They answer some open questions proposed in Xin et at.(2022) and also provide experiments on a three-task learning problem to verify the theoretical results.

**Strengths:**

- It is a novel and interesting problem to study whether linear scalarization can fully explore the Pareto front. The authors provide necessary and sufficient conditions for this in the under-parametrized regime, which is the main contribution of this paper.
- The notations and presentations in this paper are clear and the proofs seem to be correct.
- They empirically study the validity of the theoretical results, and the experiments are reasonable, and the results are convincing to me.

**Weaknesses:**

- The authors study only two cases (q = 1 and q = k - 1) in the under-parametrized regime, but are these two extreme cases representative for other cases in the under-parametrized regime?
- This paper discusses linear scalarization in the under-parametrized regime. However, over-parameterization is common in deep learning, where the network has sufficient capacity to adapt to the target tasks. Does this mean that linear scalarization in the overparameterized regime does not have the fundamental limitations discussed in this paper?
- I commend the authors for attempting to answer some open questions in Xin et al. (2022), but it lacks an explanation for why SMTOs have no improvement over linear scalarization in deep multitask learning, which is found in Xin et al. (2022).

**Questions:**

Please see the weaknesses.

**Limitations:**

The theoretical analysis in this paper is only applied to the linear network and under-parametrized regime, which is a limitation of this paper.

---

> ### Author Rebuttal · Authors · 2023-08-10
>
> We would like to thank Reviewer 2hHW for taking the time to review our paper. We appreciate that they found our work interesting and novel. Below we attempt to answer the reviewer’s questions, following the order they were made.
>
> **Are $q=1$ and $q=k-1$ representative?** A rigorous study for the general under-parametrized regime is challenging, primarily due to the hardness of characterizing the feasible region. As demonstrated in Section 3, our analysis crucially relies on the algebraic form of the surfaces, but we are unable to obtain a closed form for general $q$. Nevertheless, we do think deeply about this question, and would like to share some of our thoughts here:
>
> - The capacity of the network (reflected by $q$) is the most important quantity in this problem, as different tasks are competing for the capacity of the network for better prediction. We note that $q=1$ and $q=k-1$ represent two extreme cases, corresponding to extreme under-parameterization (least capacity) and mild under-parametrization (close-to-optimal capacity). Since the results for these two cases are similar, plus it is reasonable to assume a *smooth* change when $q$ varies, we expect the same conclusions to hold for the $q$'s in between.
>
> - We make a conjecture regarding the general conclusion. We hypothesize there is a function $F$ that transforms the gram matrix to its inverse as we increase $q$. Specifically, $F$ takes $q$ as input and outputs a matrix, $F(1)$ equals the gram matrix, and $F(q-1)$ equals the inverse of the gram matrix. The necessary and sufficient condition of full-exploration for general $q$ will then be: '$F(q)$ is doubly non-negative'.
>
> - Finally, we strongly believe that the multi-surface structure that we have revealed is both universal and fundamental. As a matter of fact, we observed a similar phenomenon when working on a toy example in a generalized setting, in which different tasks have different input $X$.  This is strong evidence showing that our conclusion can have broad applicability.
>
> **The under-parametrized regime.** Please refer to the first paragraph in point 1 of the general response. In short, while the over-parameterized regime is indeed more practical, we find it important to identify *certain* problem settings, in which we can demonstrate the weakness of linear scalarization while revealing potential benefits of SMTO methods. This helps to counter the recent arguments in the field which suggest linear scalarization is sufficient for MTL, and encourages further research in developing novel SMTO methods.
>
> **Comparison with Xin et al. (2022).** Please refer to the second paragraph in point 1 of the general response. In short, the results in these two works are not directly comparable and do not necessarily contradict each other; instead, they complement each other by providing a more complete view on the strengths and weaknesses of linear scalarization and SMTO methods.

---

> > ### Comment · Reviewer_2hHW · 2023-08-14
> >
> > Thanks for the response, I believe it is an interesting paper, and I still have some concerns:
> >
> > 1.  Is the under-parametrized regime commonly used in recent works on multi-task learning? Is it meaningful to show that linear scalarization is not sufficient in this regime?
> >
> > 2. The authors study only two cases ($q = 1$ and $q = k-1$) in the underparametrized regime. I understand that it is difficult to study the case where $q$ is not equal to $1$ or $k-1$ in general, so the authors assume that the result changes slightly when $q$ varies. However, can you state this assumption explicitly in your theoretical analysis? Why is this assumption reasonable?

---

> > > ### Author Response · Authors · 2023-08-14
> > >
> > > Thank you for the response.
> > >
> > > Regarding your first concern, we have two comments:
> > > - From a theoretical perspective, studying the under-parametrized regime is necessary for linear MTL. Otherwise, the model will have sufficient capacity to fit all tasks perfectly, and there is essentially no conflict between different tasks. In other words, it is only within the under-parametrized regime that the 'competence among difference tasks' can be accurately reflected, which we believe to be a core of MTL. We refer to the reviewer to a prominent work [1], whose main result (Proposition 3) is derived within the under-parametrized regime.
> > > - On the empirical side, it is pointed out in the literature (see [2]) that under-parameterized models are actually favorable in MTL, as they can help with information transfer and lead to better generalization.
> > >
> > > Based on the above points, we believe our study will be of interest to both theorists and practitioners in MTL.
> > >
> > > Regarding your second concern, we will elaborate further on our previous response:
> > > - We believe a larger $q$ (which implies larger model capacity) should intuitively help in our problem. We have two observations to support this argument: 1) when the model has sufficient capacity to fit all tasks (i.e., $q \ge k$), the Pareto front reduces to a singleton and can be explored by scalarization; 2) the necessary and sufficient condition under $q=1$ ($C_1$) is more restrictive than the condition under $q=k-1$ ($C_{q-1}$). Specifically, under the simplified probabilistic model discussed in Section 3.3, we can prove that the probability such that $C_1$ holds is strictly smaller than that of $C_{q-1}$.
> > > - Therefore, for the case of general $q$, we expect the necessary and sufficient condition to be more restrictive than $C_{q-1}$ while being less restrictive compared to $C_1$. Since $C_{q-1}$ still does not hold in general, we expect the hardness of full exploration to be true for all $q \in [1,q-1]$.
> > > - Finally, we believe it is more important to reveal 'why' scalarization fails in terms of full exploration. Our key observation is that the multi-surface structure of the feasible region, along with the associated phenomenon of 'gradient disagreement', leads to the failure of full exploration. While it may not be obvious that the conclusion should change smoothly with $q$, we believe it is reasonable to expect a smooth change of the feasible region from a geometric perspective (i.e., that the feasible region will not abruptly collapse into a single surface as we increase $q$). This hypothesis, combined with our observation that gradient disagreement leads to the failure of full exploration, yields the conclusion for general $q$. We leave the rigorous analysis to future work.
> > >
> > > Again, we would like to thank Reviewer 2hHW for engaging in the discussion. We will make sure that the above points are adequately addressed in the revision, and will be happy to answer further questions.
> > >
> > >
> > > References
> > >
> > > [1] Wu, Sen, Hongyang R. Zhang, and Christopher Ré. "Understanding and improving information transfer in multi-task learning." In ICLR 2023
> > >
> > > [2] Wang et al., "Can Small Heads Help? Understanding and Improving Multi-Task Generalization". In WWW 2022

---

> > > > ### Comment · Reviewer_2hHW · 2023-08-15
> > > >
> > > > Thank you for your detailed response, which addresses most of my concerns. I will raise my score accordingly.

---

> > > > > ### Author Response · Authors · 2023-08-15
> > > > >
> > > > > Thank you for the time and effort in reviewing our paper!

---

### Official Review · Reviewer_c3Pi · 2023-07-08

**Soundness:** 3 good
**Presentation:** 3 good
**Contribution:** 2 fair
**Rating:** 5
**Confidence:** 3

**Summary:**

This paper studies the linear scalarization approach in multi-task learning (MTL). It shows theoretically that the linear scalarization is not able to fully capture the Pareto optimal (PO) solutions. It also identifies necessary and sufficient conditions of full exploration of the PO solutions for under-parameterized two-layer linear MTL models.
This explains why scalarization fails in certain cases. Experiments are performed to further verify the theoretical findings.

**Strengths:**

1. The paper studies an important and timely problem, focusing on whether the popular approach -- linear scalarization for MTL are able to fully capture the Pareto front.

2. The perspective of this paper and techniques to develop the theory under the specific two-layer linear models are unique.

3. Paper is well written and easy to follow.

**Weaknesses:**

1. There is a lack of discussion of related works that also identifies the failure cases of linear scalarization fully capturing the Pareto front. See [1,2,3,4].
What is the unique contribution of this paper compared to these prior works? Some discussions need to be provided to distinguish with the prior works.

---
[1] Lin, Jiguan G. “Three Methods for Determining Pareto-Optimal Solutions of Multiple-Objective Problems”

[2] Zadeh LA, "Optimality and non-scalar-valued performance criteria"

[3] Goicoechea A, Hansen DR, Duckstein L, "Multiobjective decision analysis with engineering and business applications".

[4] R. Timothy Marler, Jasbir S. Arora, "The weighted sum method for multi-objective optimization: new insights"

[5] Bo Liu, Xingchao Liu, Xiaojie Jin, Peter Stone, Qiang Liu, "Conflict-Averse Gradient Descent for Multi-task Learning"

---

2. In line 56-57, it is too restrictive to claim that "if scalarization cannot fully explore the Pareto front, there is no inherent advantage of SMTOs over scalarization." In fact, there are other benefits of SMTOs beyond this. For SMTO methods like MGDA, the motivation is to find the steepest common descent direction for all objectives at each iteration. And the benefit sometimes lies in the optimization procedure besides the solutions they find. See the toy example in [5]

3. Even though the paper shows that linear scalarization approach is not able to fully explore the Pareto set in some settings, there is no discussion on whether SMTOs are able to fully explore the Pareto set in the same settings.

4. Some questions need to be clarified. See **Questions**



-----------------------------update post rebuttal-------------------------------

I have read the rebuttal and participated in the discussion.

The rebuttal partially addressed my concerns so I remained relatively positive about this work.

It could further improve with a more thorough discussion of the classical works and more precise positioning of their contributions.

--------------------------------------------------------------------------------------

**Questions:**

1. I do not agree with the claimed contribution 2, line 63-65, where the authors claimed to give the "first guarantee for full exploration in the presence of non-convexity."

- Please see reference [1], where necessary conditions of Pareto optimality are given for the linear scalarization method with "p-directionally convex" requirement of the criterion space, which is not convex.

- Please see reference [2,3], where sufficient conditions of (weak) Pareto optimality are given for the linear scalarization method.

- Please discuss the difference and relations with the listed prior works. Perhaps you need to add the specific settings, such as "for under-parameterized two-layer linear networks".

2. Since this paper is focused on the underparameterized model with $q < k$, how does the theory in this paper helps understanding the question posed in Xin et al. (2022), which is for deep models usually overparameterized?

3. What are the implementation details for visualization in Figure 1?

4. In line 170, what does "gradients disagree" mean? Does it mean they are not in the exactly same direction or the angle between the two gradients is larger than a threshold? There should be a formal definition.



---
[1] Lin, Jiguan G. “Three Methods for Determining Pareto-Optimal Solutions of Multiple-Objective Problems”

[2] Zadeh LA, "Optimality and non-scalar-valued performance criteria"

[3] Goicoechea A, Hansen DR, Duckstein L, "Multiobjective decision analysis with engineering and business applications".



---

### Minor

Typos and grammar errors:

Line 2: "since its inception" -> "because of its inception"
Line 96: "vecotrs" -> "vectors"

**Limitations:**

The authors have adequately addressed the limitations in Section 5.

I expect the authors to conduct a thorough review with the most relevant works, see Weaknesses-1.

Overall, this paper studies an important problem with a unique perspective.
I am willing to increase my score if the authors can address these questions.

---

> ### Author Rebuttal · Authors · 2023-08-10
>
> We express our sincere gratitude to Reviewer c3Pi for taking the time to review our paper, providing valuable and constructive feedback, and pointing out useful references. We hope to address all comments in the review below, following the order they were made.
>
> **Comparison with prior works (Weakness 1 and Question 1).** Thank you for pointing out the references! After carefully checking these papers, we find that [1] does provide a sufficient condition (p-directionally convex) for full-exploration (rather than a necessary condition), which we were previously unaware of. We are grateful to the reviewer for pointing this out and will for sure 1) incorporate it in the related work; 2) revise the second point of contribution accordingly. However, we are unable to find necessary or sufficient conditions for full-exploration in [2,3,4]. We kindly request the reviewer to further refer to the relevant content in these papers.
>
> One obvious difference between our work and prior works, as the reviewer has mentioned, is that we focus on a specific setting with a concrete model (linear networks), while prior works provide more general (and therefore much weaker) results, concerning the properties of the objective functions. But we believe the most important difference that allows our work to stand out is that the conditions we uncover are **both necessary and sufficient**, so they provide a complete picture as to when scalarization can fully explore the PF in our setting. Importantly, this facilitates the understanding of the *hardness* of full-exploration. By examining the restrictiveness of the condition, we come to the conclusion that such weakness of scalarization is fundamental. In contrast, to the best of our knowledge, there is no prior work that identifies a both necessary and sufficient condition in any non-convex setting. A sufficient condition alone is not as indicative of the hardness of full-exploration or the fundamental weakness of scalarization.
>
> **Overclaiming (Weakness 2).** We agree with the reviewer's comment and will revise this sentence. We understand that there are other aspects in comparing scalarization and SMTOs aside from the one we take, and have pointed out in the future direction (Line 356-357) that a rigorous analysis of the advantages of SMTOs is an important future work.
>
> **Advantages of SMTO methods (Weakness 3).** We agree this is a valid point, and have replied in point 4 of the general response. In short, we don’t think one SMTO method can fix the issue of linear scalarization in every scenario and we were not advocating for one particular method such as MGDA. On the other hand, we do believe that developing novel SMTO methods is a valuable line of research as they provide great flexibility in exploring different parts of the feasible region, and the argument that linear scalarization is sufficient for MTL should be rejected. We also perform additional experiments on MGDA and its variants to showcase that their capabilities of finding balanced solutions are not affected by the choice of random seed (see Figure C in the attached PDF in the general response).
>
> **The under-parametrized regime (Question 2).** Please refer to point 1 of the general response. In short, our results do not have direct implications on the results of Xin et al. (2022), as they are established under different settings. Rather, they complement each other by providing a more complete view on the strengths and weaknesses of linear scalarization and SMTO methods.
>
> **Implementation details of Figure 1 (Question 3).** Figure 1 is generated from a simple three-task linear MTL problem that we constructed, using Eq. (4). Specifically, we set $\hat y_1\approx(0.98,0,0.2), \hat y_2\approx(-0.49,-0.85,0.2), \hat y_3\approx(-0.49,0.85,0.2)$ (the number of data points is $n=3$; this is a rotated version of the equiangular tight frame), set $q=1$ (the width of the network is one, i.e., under-parameterized), and plotted the achievable points of Eq. (4) by sweeping $P_Z$ (the set of rank-1 projection matrices). The software we used is Mathematica. We will include the details in the Appendix.
>
> **Gradient disagreement (Question 4).** We are sorry for the confusion. ‘Gradient disagreement’ refers to a situation where a point lies at the intersection of two surfaces, and the tangent planes to each surface at that point are different. Equivalently, it means that the two gradients are not in the exactly same direction, so your first hypothesis is correct. We will formally define this concept in the revision. The reviewer can also refer to point 3 of the general response for more explanations.
>
> **Minor.** Thanks for pointing them out, we will revise accordingly.
>
>
> **References**
>
> [1] Lin, Jiguan G. "Three methods for determining Pareto-optimal solutions of multiple-objective problems." Directions in Large-Scale Systems: Many-Person Optimization and Decentralized Control. Boston, MA: Springer US, 1976. 117-138.
>
> [2] Zadeh, Lofti. "Optimality and non-scalar-valued performance criteria." IEEE transactions on Automatic Control 8.1 (1963): 59-60.
>
> [3] Goicoechea, Ambrose, Don R. Hansen, and Lucien Duckstein. "Multiobjective decision analysis with engineering and business applications." (No Title) (1982).
>
> [4] Marler, R. Timothy, and Jasbir S. Arora. "The weighted sum method for multi-objective optimization: new insights." Structural and multidisciplinary optimization 41 (2010): 853-862.

---

> > ### Comment · Reviewer_c3Pi · 2023-08-17
> > **Discussion of related works**
> >
> > Thanks for the rebuttal.
> >
> > Regarding the concern of **Comparison with prior works (Weakness 1 and Question 1)**, there seems to be some confusion in definitions.
> >
> > I will first clarify some definitions of **sufficient and necessary conditions for (weak) Pareto optimality (PO)** that are used in prior works.
> > For easy reference,  there is a survey paper [5, Section 2.3] that provides all the definitions.
> >
> > 1. If a formulation (e.g. linear scalarization) provides a **sufficient condition for PO**, its solutions (given all possible scalarizations) are always Pareto optimal, though it may not cover all Pareto optimal points. The solution set is a subset of the Pareto optimal set.
> >
> > 2. If a formulation provides a **necessary condition for PO**, then a Pareto optimal point must be a solution to the formulation, though some solutions may not be Pareto optimal. The Pareto optimal set is a subset of the solution set.
> >
> > To ensure linear scalarization fully explores the Pareto front (sufficient condition for full exploration), it means the PO set is a subset of the solutions of linear scalarization, the **necessary condition for PO**.
> >
> > [1] provides necessary conditions of PO, or sufficient condition for full exploration, and [2,3] provide sufficient conditions for PO.
> >
> > So I think [2,3] is not the same as the condition studied in this paper. Thank you for your response! And it would be better if you can have more discussion on these prior works.
> >
> > ---
> >
> > [5] R.T. Marler and J.S. Arora, "Survey of multi-objective optimization methods for engineering"

---

> > > ### Author Response · Authors · 2023-08-18
> > >
> > > Thank you for the clarification. We now have a better understanding of what "sufficient/necessary conditions for PO" mean. Apart from [1] which was already included, we will further add these results to the related work section---the discussion below Eq. (3) in [2], Theorem 1 in [5], and the discussion below Eq. (4.60) in [6] (another sufficient condition for PO we are aware of). Unfortunately, we failed to access [3] online; we appreciate it if the reviewer could point us to an accessible source. We thank the reviewer once more for bringing these classical results to our attention, and will make sure to give proper credits to them in our revision.
> > >
> > > Among the vast literature that concerns Pareto optimality in multi-task learning, one thing that we find missing is a systematic and rigorous study of *when and why scalarization fails to achieve a specific Pareto optimum*. Prior work typically resorts to *hypothetical examples* (e.g., Figure 4.9 in [6]), or applies intuitive yet theoretically-unjustified descriptions (e.g., "the non-convex regions of Pareto fronts" in [Zhang 2023](https://openreview.net/pdf?id=M8rwWdaGa6x)) to demonstrate the failure modes of scalarization. Our study---identifying the multi-surface structure of the feasible region and uncovering the phenomenon of gradient disagreement---serves as a first step towards filling this important gap in the literature. We hope this further underscores the significance of our work, beyond providing a more balanced and comprehensive view on the strengths and weaknesses of scalarization and SMTO.
> > >
> > > Reference
> > >
> > > [6] Boyd, Stephen P., and Lieven Vandenberghe. Convex optimization. Cambridge university press, 2004.

---

### Official Review · Reviewer_J52d · 2023-07-09

**Soundness:** 3 good
**Presentation:** 3 good
**Contribution:** 3 good
**Rating:** 6
**Confidence:** 2

**Summary:**

This paper revisits linear scalarization in multi-task learning from a theoretical perspective.
The authors reveal that scalarization is, in general, incapable of tracing out the Pareto front.
Specifically, when the model is under-parametrized, a multi-surface structure of the feasible region is revealed by the authors and it can be used to identify necessary and sufficient conditions for full exploration.
Experimental results further verify the theoretical findings.

**Strengths:**

1. This paper provides rigorous analysis to answer an important question in multi-task learning: whether linear scalarization is capable of
fully exploring the Pareto front?
The authors provide insightful remarks for the theories, and the figures provide intuitive explanations for the multi-surface structure of the feasible region.

2. The paper is well-written and easy to follow. Sufficient preliminaries and explanations are provided.

**Weaknesses:**

1. The paper only studies the under-parametrized regime. It remains unclear whether the conclusion still holds for over-parametrized regimes where the model capacity is large enough and task competence does not exist.

2. The paper mainly focuses on a two-layer neural network. Therefore, it remains unclear whether the conclusion still holds for non-convex functions.

**Questions:**

Please check the weakness.

**Limitations:**

Yes.

---

> ### Author Rebuttal · Authors · 2023-08-10
>
> We would like to thank Reviewer J52d for taking the time to review our paper. We are grateful that the reviewer appreciates our theoretical contributions. The concerns raised by the reviewer are addressed in the general response. Specifically,
>
> **The under-parameterized regime.** Please refer to point 1 of the general response. In short, while the over-parameterized regime is indeed more practical, and that our results might not directly transfer, we still find it important to identify *certain* problem settings, in which we can demonstrate the weakness of linear scalarization while revealing potential benefits of SMTO methods. This helps to counter the recent arguments in the field which suggest linear scalarization is sufficient for MTL, and encourages further research in developing novel SMTO methods.
>
> **Toy models.** Please refer to point 2 of the general response. We emphasize that even for two-layer linear networks, the loss function is *not* convex w.r.t. the model parameters, and that both the techniques and results in this paper are novel (to the best of our knowledge) and greatly advance those in the literature of convex analysis.

---

> > ### Comment · Reviewer_J52d · 2023-08-18
> >
> > Thank the authors for addressing my concerns. Although the results may not hold in the over-parameterized regime, this paper still provides some new insights into the weakness of linear scalarization and the strength of SMTO methods. I prefer to increase my score.

---

> > > ### Author Response · Authors · 2023-08-18
> > >
> > > Thanks for your acknowledgment of our response and taking the time to review our paper!

---

### Official Review · Reviewer_nBcV · 2023-07-18

**Soundness:** 2 fair
**Presentation:** 2 fair
**Contribution:** 2 fair
**Rating:** 5
**Confidence:** 4

**Summary:**

In this paper, the authors introduce a novel mathematical framework to answer the problem “Under what conditions can (or cannot) linear scalarization recover the Pareto front of a given multi-objective optimization problem?”. To this end, the paper considers the multi-task learning setting  using a simple linear multi-task learning model in the under-parameterized regime, and introduce new concepts like “feasible regions” and "gradient disagreement", and state that the intersection points of aforementioned feasible regions are unattainable by optimizing any linear scalarization of the multiple objectives. Accordingly, the paper derives necessary and sufficient conditions as to when optimizing any linear scalarization of the multiple objectives can recover the Pareto front by checking for situations where the Pareto front of the problem lies entirely in one so called feasible region. The paper then provides some empirical validation of the theory that suggest linear scalarization indeed fail to cover the Pareto front, especially the “balanced” Pareto optimal points. The empirical results further suggest that Specialized Multi-Task Optimizers (SMTOs) like MGDA and MGDA-UB can achieve a more “balanced” Pareto optimal points compared to linear scalarization.

**Strengths:**

* The paper is aimed at finding necessary and sufficient conditions for when scalarization can not recover the Pareto front, which is an important and pertinent problem in the field of multi-objective optimization.

* This paper provides a mathematical framework for analyzing the aforementioned problem, which may provide insights into how to select a method of solution for a multi-objective optimization problem based on the nature of the problem.

**Weaknesses:**

* The necessary and sufficient conditions derived for linear scalarization to recover the Pareto front in this paper requires Pareto optimal objective vectors to be “on the same feasible surface”, rather than being at an “intersection point of feasible surfaces”, as described in  Lemmas 3.4, 3.5. Yet, it is unclear why these intersection points cannot be attained by some scalarization of the objectives. Specifically, the “gradient disagreement” concept is unclear; for example how to mathematically characterize this gradient in the objective space, and why this gradient is related to the optimization of the multiple objectives.

* It is unclear what is the origin ((0, 0, 0) point ) of the objective spaces used in the Figures 1 and 3. Thus, it is a bit hard to understand the provided illustrations of feasible regions in the objective space.

* While the paper provide some weakness (i.e. the inability to recover the whole Pareto front of a problem) of scalarization in the under-parameterized regime, it does not give theoretical validation whether SMTOs can overcome this weakness. The empirical validation for showing that SMTOs can overcome the aforementioned weakness have some limitations, as described in following points. Thus, the conclusion of the paper seems to be not justified from the theoretical and empirical results.

* The experiments done to verify theory seems inconsistent for SMTOs and linear scalarization methods. Specifically, SMTOs are optimized using iterative methods for 100 epochs, while scalarization solutions are obtained from the corresponding closed form expression for the optimum solution. A fair comparison of methods would be either running SMTO methods for a very large number of iterations, or using an iterative updating scheme for scalarizations for the same number of epochs.

* By nature of the MGDA algorithm, if deterministic gradients are used, the algorithm should terminate at the first encounter of a Pareto stationary point (a notion of stationarity for multiple objectives similar to stationarity in single objective optimization), which is often a point either at least one objective have achieved stationarity, or the point where the objectives begin to conflict with one another (see some discussion on this here [1]). Thus, this kind of point is usually a point that will favor one objective over the others (unless the objectives are perfectly aligned). In this sense, it is unclear how MGDA have achieved an interior point of the Pareto front, as implied by Figure 3.

* At the beginning of the paper (such as in the abstract) the authors pose the weakness of scalarization over SMTOs as the inability to “fully recover” the Pareto front, while the empirical evaluations suggests that SMTOs can also only recover a single Pareto optimal point, which seems to not align with the initial message.

Minor comments

* Definition of Pareto front introduced in the paper seems not standard (line 119). Specifically, in this paper it is referred to as the collection of  Pareto optimal points (PO) $\theta^*$, while usually Pareto front is the set of objective values $\\{L_i(\theta^*)\\}_{i\in[k]}$  corresponding to Pareto optimal points.

[1] Liu, Xingchao, Xin Tong, and Qiang Liu. "Profiling pareto front with multi-objective stein variational gradient descent." Advances in Neural Information Processing Systems 34 (2021): 14721-14733.

Edit: Added missing reference.

**Questions:**

* What is the mathematical definition of the gradients depicted in Figure 3, and what is the precise relationship between the “gradient disagreement” and the inability to recover this point by optimizing a linear scalarization of the objectives?

* Could the authors provide some intuition or some physical meaning of the feasible regions described in the paper, and how they relate to the multiple objectives?

* Would an implementation change of the experiments as described in the previous section (point 4) change the solution distribution of each algorithm compared to Figure 3?

* Given the nature of MGDA algorithm as described in previous section (point 5), could the authors provide the reason for the convergence of MGDA like algorithms for the interior point of the Pareto front of the problem?

**Limitations:**

Limitations of the proposed framework are discussed discussed in the paper. In addition to these limitations, the paper does have limited validation as to SMTOs can overome the weaknesses of linear scalarization, as described in previous sections.

---

> ### Author Rebuttal · Authors · 2023-08-10
>
> We thank Reviewer nBcV for their insightful comments and constructive feedback. We address concerns and answer questions posed by the reviewer below, following the order they were made.
>
> **Why scalarization cannot explore intersection points (Weakness 1 and Question 1).** Please refer to point 3 of the general response.
>
> **The origin point in the figures (Weakness 2).** Figures 1 and 3 are generated from MTL problems with three tasks, and the feasible regions indicate the achievable set of MSEs (each axis corresponds to the MSE on a task; the smaller the better). Hence, the origin point (0,0,0) indicates solving the MTL problem perfectly, whereas the point (1,1,1) corresponds to a bad solution and is not Pareto optimal. We will regenerate Figure 3 to ensure consistency with Figure 1 by setting the origin (0,0,0) on the front and the point (1,1,1) on the back.
>
> **Advantages of SMTO methods (Weakness 3).** We agree this is a valid point, and have replied in point 4 of the general response. In short, we don’t think one SMTO method can fix the issue of scalarization and we were not advocating for any specific method. On the other hand, we do believe that developing novel SMTO methods is a valuable line of research as they provide flexibility in exploring different parts of the feasible region, and the prevalent argument that scalarization is sufficient for MTL does not hold in all the cases, at least in the under-parametrized regimes. We also perform additional experiments on MGDA and its variants to show that their capabilities of finding balanced solutions are not affected by the choice of random seed (see Figure C in the attached PDF in the general response).
>
> **Unfair comparison (Weakness 4 and Question 3).** We respectfully disagree with this point since 1) using closed-form solutions will only favor scalarization. Therefore, if our conclusion is valid in the current setup, it’s going to hold with finite iterations as well; 2) in this paper, we study the weakness of scalarization on the *representation* level. Using finite iteration methods without resorting to the existing closed-form solutions will unnecessarily introduce artifacts from the *optimization* procedure.
>
> **Why does MGDA converge to interior points? (Weakness 5 and Question 4)** After checking the reference, we failed to find a precise statement/theorem that claims or implies that MGDA will converge to a point on the periphery of the Pareto front. We would appreciate it if the reviewer could elaborate further on this point, or point to the precise location in the reference.
>
> To further strengthen our empirical findings, we perform some additional experiments. First, we plot the optimization trajectories of MGDA and MGDA-UB in Figure A in the attached PDF. Both algorithms first overfit task 2 and approach scalarization optimal points, but then turn sharply twice and finally reach more balanced solutions. Second, Figure C demonstrates that the balancedness of the final solution is not affected by the random seed.
>
> **Deviation from the initial message (Weakness 6).** We apologize for the confusion here. We did not intend to show that a particular SMTO method can overcome the weakness of scalarization. Instead, here are some high-level messages that we would like to deliver:
> - Xin et al. (2022) observed that scalarization is capable of tracing out the PF when the objective functions are convex, and used it to support their hypothesis that there is no inherent advantage of SMTO methods and scalarization is sufficient for MTL. Our theoretical analysis demonstrates that the reasoning is untenable. Specifically, even with some mild non-convexity, scalarization is incapable of fully exploring the PF. This calls for further research on explaining the empirical success of scalarization and reconciling with the theoretical limit that we have revealed.
> - For the experiment on MGDA, we did not intend to show that it can overcome the limitation of scalarization. As a matter of fact, we do not think there exists a single SMTO method that can fully explore the PF in every scenario and can be taken as the gold standard in MTL. Rather, we would like to demonstrate that certain parts of the PF, which are not achievable by scalarization yet are of potential interest (e.g., a region that contains more balanced solutions), can be achieved by certain SMTO methods. This implies that research on SMTO is not futile, and helps to reject the claim that scalarization is sufficient for MTL. In doing so, we hope to foster a healthier and more balanced development in the field of MTL.
>
> **Explanation of the feasible region (Question 2).** Feasible region is originally defined as the set of all possible points in $\mathbb{R}^k$ (each dimension is a quadratic function, as shown in Eq. (4)) that can be achieved by varying $P_Z$, a projection matrix determined by the weight of the hidden layer $W$. Intuitively, the feasible region is a reflection of the network's representation power. In our work, we slightly modify the definition of feasible region by imposing some restrictions on $P_Z$ (see Eq. (5)). This helps to simplify the subsequent analysis, while keeping the PF intact, meaning that the original PF is still a subset of our defined feasible region.
>
> The multi-surface structure arose when we were performing a fine-grained analysis of the feasible region. A possible hypothesis is related to game theory—viewing each task as a player, the number of surfaces equals the possible coalitions formed by the $k$ players ($2^k$). Each surface corresponds to one particular coalition and represents a set of possible outcomes the coalition can obtain, potentially favorable to the players within. Finally, the intersection of different surfaces indicates conflict among coalitions.

---

> > ### Comment · Reviewer_nBcV · 2023-08-15
> >
> > I thank the authors for the clarifications, specially on the concepts of the feasible regions and the gradient disagreement. I have few follow up questions and comments based on these responses.
> >
> > * In the 2D objective space figure provided in the link, is it possible to point out the corresponding feasible regions and the corresponding intersection point of these feasible regions?
> > * As mentioned in point 4 of the response, did the authors consider “artifacts of optimization procedure” might also affect (possibly in a favorable way) the SMTO methods?
> > * In the caption of Figure C in pdf, it says “Both algorithms tend to find solutions located in the interior of the feasible region”? Does this mean these are not necessarily in the Pareto front?
> > * Regarding point 5 of the response,  the reference is not for some theoretical results on why MGDA can’t converge to the interior point on Pareto front, rather for empirical results, similar to what this paper is trying to provide. Some of the locations in the paper that discuss and demonstrate this issue are: Figure 1 and Example 1 (a simple toy example), Figure 2 (b) (stochastic version of MGDA, applied for several datasets).
> > * I feel the setting used in this paper is significantly  incomparable to that of Xin et al. (2022) to imply that results in this paper can dispute the claims in Xin et al. (2022), as suggested in point 6.

---

> > > ### Author Response · Authors · 2023-08-15
> > >
> > > Thanks for your response. We will answer your questions in a point-by-point manner:
> > >
> > > - **Feasible region.** As stated in the caption of the figure, the shaded region (denoted as $\mathcal{O}$) is the set of achievable values. Rigorously speaking, the feasible region that we have defined is a subset of $\mathcal{O}$, but these two can be treated equally in our setting. The reason is that the feasible region contains the entire Pareto front (PF) (explained at the beginning of Section 3.1), and since our interest lies in the exploration of PF, ignoring the points that are Pareto dominated will not affect our analysis.
> > >
> > > - **Intersection point.** The figure comes from a classical textbook [1] and demonstrates a *hypothetical* failure case of scalarization. In contrast, the intersection point is a new discovery in a concrete setting and hence cannot be reflected in the figure. As stated in the paper, what we have done partly is to 'penetrate through the surface and gain a holistic view of the feasible region'. By solely examining the surface, it is not possible to identify the intersection points or the phenomenon of gradient disagreement. We believe this is one of our core contributions: our work advances previous studies by offering new insights into the failure modes of scalarization, going beyond mere hypothetical examples.
> > >
> > > - **Artifacts of optimization procedure.** This is not likely, especially in light of the additional experiments that we have performed. Specifically, we stick to the original implementation of MGDA and MGDA-UB with no modifications at all. To eliminate the effect of random initialization, we performed additional experiments and observed similar results (see Figure C of the attached PDF). Finally, we observed consistent results when varying the tasks, further consolidating the effectiveness of MGDA and MGDA-UB in our setting.
> > >
> > > - **Interior point.** We are sorry for the confusion. We intend to claim that the two algorithms tend to find solutions located at the interior of the PF, which is a part of the *surface* of the feasible region. As stated previously, the feasible region contains the entire PF, and the PF must lie on the surface of the feasible region. Furthermore, MGDA and MGDA-UB are guaranteed to converge to Pareto optimal solutions (Theorem 2.2 of [2] and Theorem 1 in [3]). Therefore, the two convergent points must lie on the surface of the feasible region and belong to the PF. In our experiments, we also plot non-negative orthants at the two convergent points and observed that they do not intersect with the feasible region, meaning that they are indeed Pareto optimum.
> > >
> > > - **Convergence of MGDA.** We agree it is possible to construct examples where MGDA converges to bad solutions. As stated in our previous response, we don't think one SMTO algorithm can work in every scenario, and the choice of algorithm should depend on the problem structure and specific requirement. Nevertheless, since our experiments are performed in a more practical scenario using real (despite simple) neural networks, and the experimental results are consistent across difference random seeds and tasks, we believe it is reasonable to claim that 'certain SMTO methods have the potential to find balanced solutions'. After all, it is not our goal to advocate for a particular algorithm; what we hope is to bolster research in SMTO and promote a healthier and more balanced developement in MTL.
> > >
> > > - **Xin et al. (2022).** We do realize that the difference of settings makes the results in these two works not directly comparable. Nevertheless, we would like to clarify a few points:
> > >   - We do not intend to refute all the claims and conclusions in Xin et al. (2022), but we do feel that there are *some* reasoning gaps in their work that we would like to point out and clarify. Specifically, it is marked bold in their paper that 'in the convex setting it is provable that no algorithm can outperform scalarization', which the authors used to set the tone for their paper. However, their experiments are performed with deep neural networks, and the authors simply connect the theory and experiments by listing a number of open questions. Through our theoretical analysis, we found that the conclusions in the convex setting do not transfer. Therefore, the theoretical results in Xin et al. (2022) cannot be used to support their empirical findings. Instead, the community needs to develop a new theory to explain the empirical success of linear scalarization.
> > >   - We see our work as a complement to Xin et al. (2022). Putting together, they provide a more comprehensive view on the strengths and weaknesses of linear scalarization and SMTO.
> > >
> > > We thank the reviewer for pointing this out and will tune down our tone in drawing comparison with Xin et al. (2022). The remaining points will also be addressed accordingly. We appreciate the reviewer's constructive feedback for our work, and will be happy to answer any further questions.

---

> > > > ### Author Response · Authors · 2023-08-15
> > > > **References**
> > > >
> > > > References
> > > >
> > > > [1] S. Boyd, S. P. Boyd, and L. Vandenberghe. Convex optimization. 2004.
> > > >
> > > > [2] Désidéri, Jean-Antoine. "Mutiple-gradient descent algorithm for multiobjective optimization." European Congress on Computational Methods in Applied Sciences and Engineering (ECCOMAS 2012). 2012.
> > > >
> > > > [3] Sener, Ozan, and Vladlen Koltun. "Multi-task learning as multi-objective optimization." Advances in neural information processing systems 31 (2018).

---

> > > > > ### Comment · Reviewer_nBcV · 2023-08-17
> > > > >
> > > > > Thank you for the additional clarification. While the paper may not exactly answer the practical issues in MTL in recent literature, I acknowledge the theoretical framework introduced in the paper is interesting, albeit the setting considered might be very restrictive. Thus, I raise my score.

---

> > > > > > ### Author Response · Authors · 2023-08-17
> > > > > >
> > > > > > Thank you for the recognition in our work, and again, for taking the time to review our paper and providing constructive feedback. We will soon compile a list of changes according to yours and the other reviewers’ suggestions in the general response.

---

### Author Rebuttal · Authors · 2023-08-10

Here we address some common concerns raised by multiple reviewers.

**1. The under-parameterized regime.** (Reviewer J52d, c3Pi, 2hHW, aGcP)

We acknowledge that the over-parameterized regime is more practical, and the conclusions in this paper may not directly generalize. As a matter of fact, even for linear over-parameterized models, scalarization can fully explore the Pareto front (PF), and the limitation that we have revealed no longer holds. Nevertheless, our main focus is to show that under *certain* problem settings, scalarization has fundamental limitations, while SMTO methods can provide benefits (e.g., finding balanced solutions). In doing so,  we strive to bolster research in SMTO by addressing and countering recent arguments found in [1,2], which excessively laud the merits of scalarization, consequently diminishing the perceived value of SMTO. Our aim is to provide a more balanced perspective on the subject and clarify the importance of SMTO in the context of relevant research.

We emphasize that we take a different perspective—the full exploration of PF—in comparing scalarization and SMTO methods than [1], which focuses on the *accuracy*. As such, the conclusions in these two works are not directly comparable. In contrast, they complement each other by providing different views on the strengths and weaknesses of these two lines of work, which we believe is of great significance to both researchers and practitioners.

**2. Simple models.** (Reviewer J52d, CFe6)

Generalization to the non-linear setting is an important future work that goes beyond the scope of our paper. Additionally,
- Even for linear networks, the objective functions are not convex w.r.t. the model parameters, so the theoretical analysis demands fundamentally different techniques than those in convex analysis (see [3]).
- As stated in the paper, our results are strong in the sense that they are inherent, i.e., our conclusions are independent of the specific optimization algorithms used to minimize the scalarization objective. Furthermore, our results can be easily generalized to multi-layer linear networks, as they have the same representation power as their two-layer counterparts.

**3. Why scalarization cannot explore intersection points?** (Reviewer nBcV, c3Pi, aGcP)

We apologize for the confusion, and provide detailed explanations in what follows.

We use Figure 4.9 in [3] for ease of illustration (see [this link](https://drive.google.com/file/d/1d-5PWLSs-MJSGpxgLr_sDdCHS2IVQpyR/view?usp=drive_link)). A point $P$ lying at the boundary of the feasible region can be achieved by scalarization, if and only if there exists a hyperplane at $P$ and the feasible region lies above the hyperplane. In Figure 4.9, $f_0(x_1)$ and $f_0(x_2)$ are positive examples as the dashed lines do not intersect with the shaded region, while $f_0(x_3)$ is a negative example. The normal vector of the hyperplane is proportional to the scalarization coefficients.

By definition, if a hyperplane at $P$ lies below the feasible region, its normal vector must be a subgradient of the surface. When the boundary of the feasible region is differentiable and the subdifferential set is non-empty, the normal vector must be the gradient of the surface, and the hyperplane becomes the tangent plane at $P$. So the mathematical characterization of ‘gradient’ would be the normal vector of the tangent plane.

Now suppose $P$ lies at the intersection of two differentiable surfaces $S_1$ and $S_2$, and that $P$ can be achieved by scalarization. Applying the above argument to $S_1$ and $P$, we know that the scalarization coefficients are proportional to the gradient w.r.t. $S_1$. Similarly, applying the above argument to $S_2$ and $P$ yields that the scalarization coefficients are proportional to the gradient w.r.t. $S_2$. This will result in a contradiction if the two gradients w.r.t. $S_1$ and $S_2$ at $P$ lie in different directions, a phenomenon we refer to as ‘gradient disagreement’.

We hope the above explanation clarifies the reviewers’ concern.

**4. Advantages of SMTO methods.** (Reviewers nBcV, c3Pi)

We acknowledge this is a limitation of our work, and have mentioned it in the future work section. Additionally,
- We hypothesize that there does not exist a *single* SMTO method that can trace out the PF in every case and therefore dominate the others. But the abundance of SMTO methods does provide flexibility for practitioners: depending on the problem structure and specific requirement, one may select proper SMTO methods that are capable of exploring *some* part of the PF. For instance, when the balancedness of solution is of utmost importance, [4] is a good fit.
- As a consequence, it is not our goal to advocate for any specific SMTO method like MGDA. Instead, by pointing out a fundamental limitation of scalarization, we hope to reject the prevalent claim that scalarization is sufficient for MTL, and bolster research in the development of novel SMTO methods. We believe our paper, combined with [1,2], provides a more comprehensive understanding of the strengths and limitations of scalarization and SMTO methods, and contributes to a healthier and more balanced understanding of different MTL paradigms.
- Finally, we conduct additional experiments on MGDA and MGDA-UB. As reflected in Figure C in the attached PDF, their capabilities of finding balanced solutions are not affected by the choice of random seed. This further strengthens our argument on the potential benefit of some SMTO methods.

**References**

[1] Xin, Derrick, et al.  Do Current Multi-Task Optimization Methods in Deep Learning Even Help? NeurIPS 2022

[2] Kurin, Vitaly, et al.  In defense of the unitary scalarization for deep multi-task learning. NeurIPS 2022

[3] Boyd, Stephen P., and Lieven Vandenberghe. Convex optimization. Cambridge university press, 2004

[4] Navon, Aviv, et al. Multi-task learning as a bargaining game. arXiv:2202.01017

---

### Author Response · Authors · 2023-08-18
**Plan of revision**

Dear reviewers,

We are grateful to all of you for taking the time to review our paper, for engaging in the subsequent discussions, as well as for providing constructive feedback. We believe the peer-review process has substantially improved the quality of our work. In what follows, we will list the changes that we intend to make on the current manuscript:
- When answering the open questions from Xin et al. (2022) in Section 3.3, we will include a paragraph to clarify that: 1) we do not intend to refute all their claims; rather, we would like to point out the reasoning gap between their theory and experiments, and call for a new theory from the research community to explain the empirical success of linear scalarization; 2) the difference in settings makes the results from these two papers not directly comparable; nevertheless, they complement each other by providing a more comprehensive view on the strengths and weaknesses of scalarization and SMTO.
- For Section 4, we will add the experiments for randomization and MGDA with different initializations. We will also clarify that 1) it is not our goal to advocate for a particular SMTO algorithm, and we don’t believe one single algorithm can work in every scenario; 2) instead, by revealing the limitation of scalarization and potentials of some SMTO algorithms, we hope to bolster the research of SMTO, to refute the claim that linear scalarization is sufficient for MTL, and to promote a healthier and more balanced development within the field of MTL.
- We will clearly define and explain the concept of gradient disagreement, and will provide detailed reasoning steps on why and how it leads to the failure of exploration at intersection points.
- We will include more discussions on the limitation of our work (mostly from a technical perspective), specifically on the under-parametrized regime and general $q$.
- We will make sure all the relevant references that the reviewers have mentioned are properly cited and discussed.
- We will address other minor points, e.g., issues with figures (implementation details and captions), overclaiming, structure of Section 3, etc.

We kindly request the reviewers to let us know if there are other changes that you would like to see in the revision.

Sincerely,

Authors of Submission 5214

---

### Decision · Program_Chairs · 2023-09-21

**Decision:**

Accept (poster)

**Comment:**

The paper is theoretically studying the linear scaling for multi-task learning. The most interesting result is the finding that no static scaling is guaranteed to find a Pareto efficient solution for the under-parameterized setting. The paper was reviewed by 6 reviewers and they all unanimously suggested acceptance. I would strongly recommend authors to incorporate reviewers' feedback and as well as add a discussion about possible extensions to over-parameterized regime.